# Accuracy of five ground heat flux empirical simulation methods in the surface energy balance-based remote sensing evapotranspiration models

Zhaofei Liu[1, 2]

[1] Institute of Geographic Sciences and Natural Resources Research, Chinese Academy of Sciences, 100101, Beijing, China.
[2] University of Chinese Academy of Sciences, 100049, Beijing, China

*Correspondence to*: Zhaofei Liu (zfliu@igsnrr.ac.cn)

**Abstract.** Based on the assessment from 230 flux site observations, intra-day and daytime ground heat flux (G) accounted for 19.2% and 28.8% of the corresponding net radiation, respectively. This indicates that G plays an important role in remote sensing (RS) energy balance based evapotranspiration (ET) models. The G empirical estimation methods had been evaluated at many individual sites, while there were relatively few multi-site evaluation studies. The accuracy of the five empirical G simulation methods in the surface energy balance-based RS ET models was evaluated using half-hourly observations. The linear coefficient (LC) method and the two methods embedded with the normalized difference vegetation index (NDVI) were able to accurately simulate a half-hourly G series at most sites. The mean and median Nash-Sutcliffe efficiency (NSE) values of all sites were generally higher than 0.50 in each half-hour period. The accuracy of each method varied significantly at different sites and at half-hour intervals. The highest accuracy was exhibited during 6:00-7:00, followed by the period of 17:00-18:00. There were 92% (211/230) sites with an NSE of the LC method greater than 0.50 at 6:30. It showed a slightly higher accuracy during night periods than during daytime periods. The lowest accuracy was observed at the period of 10:00-15:30. The sites with an NSE exceeding 0.50 only accounted for 51% (118/230) and 43% (100/230) at 10:30 and 13:30, respectively. The accuracy of the model was generally higher in Northern Hemisphere sites than in Southern Hemisphere sites. In general, the highest and lowest accuracies were observed at the high- and low-latitude sites, respectively. The performance of the LC method and the methods embedded with NDVI were generally satisfactory at the Eurasian and North American sites, with the NSE values of most sites exceeding 0.70. Conversely, it exhibited relatively poor performance at the African, South American, and Oceanian sites, especially the African sites. Both the temporal and spatial distributions of the accuracy of the G simulation were positively correlated with the correlation between G and the net radiation. Although the G simulation methods accurately simulated the G series at most sites and time periods, their performance was poor at some sites and time periods. The application of RS ET datasets covering these sites requires caution. Further improvement of G simulations at these sites and time periods is recommended for the RS ET modelers. In addition, variable parameters are recommended in empirical methods of G simulation to improve accuracy. Instead of the Rn, finding another variable that has a physical connection and strong correlation with the G might be a more efficient solution for the improvement, since the weak correlation between the G and Rn is the main reason for the poor performance at these regions.

## 1 Introduction

Accurate simulations of evapotranspiration (ET) represent the core of hydrological processes, crop growth, and ecosystem water efficiency simulations (Ponce-Campos et al., 2013). Remote sensing (RS) is the only viable technique that can provide relatively frequent and spatially contiguous ET measurements on global and regional scales (Zhang et al., 2016; Ait Hssaine et al., 2020). Surface energy balance is the main method used in RS ET modeling (Zhang et al., 2016; Chen and Liu, 2020). Ground heat flux (G) accounts for a significant fraction of the surface energy balance (Pauwels and Daly, 2016), but there is insufficient research on these models compared with sensible heat flux (H) (Mohan et al., 2020; Wu et al., 2020). Over bare soils or sparsely vegetated surfaces, G can reach half of the net radiation (Rn) (Heusinkveld et al., 2004). Even under full vegetation cover, G is significant, especially when turbulent processes are less active (Gentine et al., 2012).

Soil heat flux is the heat flux occurring in a layer of soil. G, which is the soil heat flux at the surface, is difficult to observe directly, due to technical limitations (Wang and Bou-Zeid, 2012; Gao et al., 2017), and direct estimation of G using RS data is not possible (Kalma et al., 2008; Allen et al., 2011; Saadi et al., 2018). Soil heat flux is generally measured using heat flux plates near the surface (within a few millimeters of the surface). This measurement is referred to as G' in this study. However, the difference between G' and G could be 50% because of the soil heat storage within the layer from the surface to the flux plate (Heusinkveld, 2004; Yue et al., 2011; Wu et al., 2020). A large error is produced if the soil heat storage is ignored in the G calculation (Meyers and Hollinger, 2004; Lu et al., 2018). In addition, the soil above the heat flux plates can lose contact with the rest of the near-surface soil matrix, adversely affecting the water and heat flow (Leuninget al., 2012; Russell et al., 2015). The spatial scale of the G observation is also much smaller than that of the H and latent heat flux (LE) estimates (Shao et al., 2008; Verhoef et al., 2012).

There are numerous schemes for estimating G (Wang and Bou-Zeid, 2012; Gao et al., 2017; Wu et al., 2020), that can be classified into two categories. According to physical mechanisms, G can be calculated from the soil heat conductivity and the vertical gradient of temperature using the so-called gradient approach (Yang and Wang, 2008). A more common approach is to combine G' at a specific reference depth with soil heat storage above (Kustas et al., 2000; Wu et al., 2020). G' can be simulated using the gradient approach or observed by heat flux plates. Soil heat storage in the soil layer above the measured depth can be calculated as the integral over the change in temperature with time multiplied by the volumetric heat capacity of the soil, which is called the calorimetry method (Liebethal and Foken, 2007; Agam et al., 2019). However, applications of these physical mechanism-based approaches are restricted to only a few sites, due to the limitations of field observations of soil thermal properties (Mayocchi and Bristowa, 1995; Kustas et al., 2000). Soil thermal properties are affected by soil texture, mineralogical composition, bulk density, and the surrounding environment (e.g., soil moisture and temperature) (Peng et al., 2017; Ju and Hu, 2018). In other words, soil thermal properties vary with time and space. In addition to these physical mechanism-based approaches, G can also be estimated using empirical methods based on Rn, H, or G' (Cellier et al., 1996; Leuning et al., 2012; Purdy et al., 2016).

To estimate ET in RS models, G is usually obtained from empirical relations with Rn. Choudhury et al. (1987) established an empirical function between G and Rn for bare and vegetated soils. They found that the ratio of G to Rn (G/Rn) was 0.4 for bare soils. For vegetated land cover, the ratio could be described by an exponential relationship with the leaf area index. Kustas and Daughtry (1990) also calibrated the ratio by ground-based measurements for bare soil, alfalfa, and cotton, and found the corresponding value to be 0.29±0.05, 0.16±0.035, and 0.27±0.02, respectively. The constant G/Rn has been used in several RS-based ET models, including the Two-Source Energy Balance (TSEB) (Norman et al., 1995), the Atmosphere-Land Exchange Inverse (ALEXI) (Anderson et al., 1997), and the Disaggregated ALEXI (DisALEXI) (Norman et al., 2003) models. The ratio values in these models ranged from 0.15 to 0.35. The Global Land Evaporation Amsterdam Model (GLEAM) used 0.05, 0.20, and 0.25, for tall canopy, short vegetation, and bare soil, respectively (Miralles et al., 2011). The G/Rn is also usually parameterized as an empirical function with the vegetation index in other RS-based ET models, including but not limited to the Surface Energy Balance Algorithm for Land (SEBAL) (Bastiaanssen et al., 1998), Surface Energy Balance System (SEBS) (Su, 2002), Mapping Evapotranspiration at high Resolution with Internalized Calibration (METRIC) (Allen et al., 2007) and Simplified TSEB (STSEB) (Sánchez et al., 2008) models. The solutions of G in the first two models were also applied to the Simplified Surface Energy Balance Index (S-SEBI) (Roerink et al., 2000) and Four-source Surface Energy Balance (SEB-4S) (Merlin et al., 2014) models, respectively. Some studies have modified the parameter values in these empirical relationships, such as the modified SEBAL (Singh et al., 2008; Faridatul et al., 2020) and TSEB models (Ait Hssaine et al., 2020). In addition, the empirical relations between G and Rn were applied to simulate G in several RS-based global ET datasets. These ET datasets include, but are not limited to, the Breathing Earth System Simulator (BESS) (Jiang and Ryu, 2016), Moderate Resolution Imaging Spectroradiometer (MODIS; MOD16A2) (Mu et al., 2011), GLEAM (Miralles et al., 2011), Numerical Terradynamic Simulation Group (NTSG) (Zhang et al., 2010) and Thermal Energy Balance (Chen et al., 2014; 2021) products. More G empirical estimation methods could be found in Sun et al. (2013) and Bonsoms and Boulet (2022).

Several studies have evaluated the empirical methods in the simulation of G. Liebethal and Foken (2007) evaluated six parameterization approaches for G by using the reference data set, in which G' at a depth of -0.2 m and the heat storage in the soil layer above -0.2 m was calculated by the gradient and calorimetry approaches, respectively. They found that the physical mechanism-based calorimetric and simple measurement approaches showed better performance than empirical methods. Similar results were also found in evaluations by Russell et al. (2015) and Gao et al. (2017) for eddy-covariance tower measurements in Idaho and Washington, respectively. Saadi et al. (2018) evaluated three empirical methods using flux station measurements and determined that the best results were obtained using the SEBAL method. However, these evaluations were limited to a single site scale because field observations of soil thermal properties were available only at a few sites. Purdy et al. (2016) evaluated six empirical methods of G simulation against G' observations at 88 flux sites. This was a very meaningful study on a global scale. However, there is a large difference between G' and G which has been described above.

The surface energy balance method provides an alternative solution for assessing the G simulation schemes (van der Tol et al., 2012). This method could avoid the inconsistent spatial scale of G with that of LE and H in field measurements. The FLUXNET dataset, which contains global flux tower observations, provides a good opportunity to evaluate G simulation methods on a global scale (Knox et al., 2019; Pastorello et al., 2020; Liu, 2021). According to the surface energy balance method, this study uses the FLUXNET dataset to comprehensively evaluate the ability of various schemes to accurately simulate half-hourly G at global flux measurement sites.

This study addresses four key objectives: (1) investigating the temporal and spatial variations and common characteristics of the empirical relationship between G and Rn; (2) evaluating the accuracy of five empirical methods in simulating half-hourly G from Rn; and (3) investigating the performance of five methods at different times during the intra-day and the spatial distribution of simulation accuracy at global flux observation sites. The evaluation results of this study are expected to provide reference for RS ET model application and developers.

## 2 Materials and methods

### 2.1 Data

This study used FLUXNET eddy covariance observations that cover all continents, including FLUXNET2015 (Pastorello et al., 2020) and FLUXNET-CH4 community products (Knox et al., 2019). FLUXNET2015 contains 212 observation sites from 1991 to 2014, while the FLUXNET-CH4 community product contains 81 sites from 2006 to 2018. The longest observational record was 25 years, whereas the shortest was less than one year. Half-hourly data series for LE, H, Rn, and G' were used. All missing values were eliminated. For example, if there were missing values on a certain day, all data on that day were discarded. Therefore, only days with fully available half-hourly data were used in the analysis. Only sites with a data series longer than 360 days were used. These eliminations ultimately meant that a total of 189 FLUXNET2015 sites and 60 FLUXNET-CH4 sites were used in the analysis because of the lack of observations (**Table S1**). There were 19 sites belonging to both FLUXNET2015 and FLUXNET-CH4. G' was not observed at 63 sites (**Table S1**). Flux observation data from four sites in Australia were obtained from the TERN OzFlux dataset. These four sites were included in FLUXNET products, but were with a longer and continuous series up to 2019 (Beringer et al., 2016). In addition, the normalized difference vegetation index (NDVI) data (Vermote et al., 2014), which were derived from surface reflectance data acquired by the Advanced Very High-Resolution Radiometer sensor, were used in this study. The dataset had a spatial resolution of $0.05 \times 0.05\,°$ and temporal coverage from June 1981 to the present.

**2.2 Methods**

**2.2.1 The surface energy balance residual method for estimating G**

The surface energy balance residual (SEBR) method (**Eq. (1)**) is an effective tool for estimating G, using the measurements of other components with a flux tower (van der Tol et al., 2012). Energy balance is an independent means of assessing G. In the surface energy balance, heat storage in the air between the ground and the height of the eddy covariance system is neglected, as is the horizontal advection of heat and other minor energy sources and sinks (Wilson et al., 2002; Leuning et al., 2012; Stoy et al., 2013). The SEBR method equation is expressed as follows:

$$G = Rn - LE - H \tag{1}$$

where G is ground heat flux; Rn is net radiation; and LE and H are latent and sensible heat flux, respectively. The estimated time series were used as referenced G for evaluating simulations of G in RS ET models. The mean and standard deviation approach was applied to detect outliers (Liu et al., 2019).

**2.2.2 Simulations of G in RS ET models**

Based on the half-hourly series of G simulated by the SEBR method at globally observed sites, the G simulation methods commonly used in five RS ET models were evaluated. The first is the linear coefficient (LC) method. It has been applied in the TSEB, ALEXI, DisALEXI, GLEAM, and other RS ET models to simulate G, but different models use different linear coefficient values. Then, the linear models of embedding the NDVI in the form of the power (Bastiaanssen, 1995) and exponential (Choudhury et al., 1987) functions (referenced as LC_NDVI_P and LC_NDVI_E) were evaluated, which were typically applied in the SEBAL (Bastiaanssen et al., 1998) and modified SEBAL (Singh et al., 2008) and SEBS (Chen et al., 2019) ET models, respectively. Finally, the linear models embedded with fractional vegetation coverage, which were applied in the SEBS and STSEB models (referenced as LC_fc_SE and LC_fc_ST), were also evaluated. LC methods embedded with the vegetation index have also been applied in many other RS ET models, such as the S-SEBI, NTSG, BESS, METRIC, MOD16A2, and SEB-4S models. However, different models may use different parameter values.

In this study, the parameters in these five methods were calibrated, rather than fixed parameter values in the original models. The half-hourly series of the observed Rn and simulated G by the SEBR method at each flux tower site were used for the optimal calibration of the parameters. These five methods are represented as LC, LC_NDVI_P, LC_NDVI_E, LC_fc_SE, and LC_fc_ST. The expressions are as follows:

$$G = \alpha\ Rn \tag{2}$$

$$G = \alpha\ [1 - 0.98\ NDVI^4]\ Rn \tag{3}$$

$$G = [\alpha\ exp(-b\ NDVI)]\ Rn \tag{4}$$

$$G = [\alpha_1 + (\alpha_2 - \alpha_1)\ (1-fc)]\ Rn \tag{5}$$

$$G = \alpha\ (1-fc)\ Rn \tag{6}$$

where α, $α_1$, $α_2$ and b are parameters to be calibrated, within the range of [0.01, 1.5], [0.01, 1.5], [0.01, 0.1], and [0.1, 1.5], respectively. At each site, daily series of each half-hour were divided into two parts: the first 80% of the data were used for parameter calibration and the rest were used for validation. The parameters of these methods were calibrated by the Nash-Sutcliffe efficiency (NSE) at each observation site.

### 2.2.3 Evaluation criteria

The simulations of G were evaluated using the G series estimated using the SEBR method for each site in each half-hourly period. The criteria tried to evaluate these simulations included the relative error (RE), root-mean-square error (RMSE), Kling-Gupta efficiency (KGE), and NSE (Gupta et al., 2009; Nash and Sutcliffe, 1970). The RE and RMSE represent the bias deviation from the observed values, whereas the KGE and NSE are indicative of the goodness-of-fit of the simulated and observed data series. The best-fit value was 1.0, and the goodness-of-fit deteriorated with increasing deviation from 1.0. The evaluation criteria were calculated as follows:

$$RE = \frac{1}{n}\sum\nolimits_{i=1}^{n} \frac{X_{si} - X_{ri}}{X_{ri}}, \tag{7}$$

$$RMSE = \sqrt{\frac{1}{n}\sum\nolimits_{i=1}^{n} [(X_{si} - \overline{X_s}) - (X_{ri} - \overline{X_r})]^2}, \tag{8}$$

$$KGE = 1 - \sqrt{(1 - CC)^2 + RE^2 + \left(1 - \frac{SD_s}{SD_r}\right)^2} \tag{9}$$

$$SD = \sqrt{\frac{1}{n-1}\sum\nolimits_{i=1}^{n} \left(X_i - \overline{X}\right)^2}, \tag{10}$$

$$NSE = 1 - \frac{\sum_{i=1}^{n}(X_{si} - X_{ri})^2}{\sum_{i=1}^{n}\left(X_{ri} - \overline{X_r}\right)^2}, \tag{11}$$

where $X_{si}$ and $X_{ri}$ are the $i^{th}$ values of the simulated and referenced G time series, respectively; $n$ is the time series length; $\overline{X_s}$ and $\overline{X_r}$ are the means of the simulated and referenced G, respectively; and $SD_s$ and $SD_r$ are the standard deviations of the simulated and referenced G, respectively.

### 3 Results

### 3.1 Intra-day distribution of observed surface energy balance items and G/Rn

The intra-day distribution characteristics of each flux variable were analyzed based on field observation data. **Figure 1** shows the intra-day distribution of half-hourly Rn, H, LE, G, and G'. The first four variables and G' were derived from the mean of 230 and 167 FLUXNET sites (**Table S1**), respectively. Overall, LE, H, and G accounted for 34.5%, 46.3%, and

19.2% of Rn, respectively. G accounted for 28.8% of Rn when only daytime periods were considered. This indicates that ignoring G in energy-based models greatly overestimates the ET values. The observed daytime G' value was only 24.1% of that of G. Considering the intra-day periods, G' was only 4.7% of G. All flux variables were stable and showed little variance from 20:00-6:00. During this period, the LE was positive and accounted for only 7% of the total daily LE, whereas other flux variables were negative. It showed a unimodal distribution for all flux variables during the day. The intra-day distribution of H showed the best agreement with the measured Rn (**Fig. 1 a**). However, the intra-day distributions of LE, G, and G' showed an overall deviation from the measured Rn. The distribution of LE and G' was generally half an hour delayed compared to the measured Rn, while that of G was half an hour earlier. The intra-day distribution of each flux variable during the daytime was compared with the Sine and Gaussian functions (**Fig. 1 b-f**). The results showed that daytime flux variables were more consistent with the latter than with the Sine function, which is commonly used to upscale instantaneous ET to daily values in RS applications. The Gaussian function perfectly matched each flux variable at any time during the day.

**Figure 1.**

The intra-day distribution characteristics of G/Rn and the ratio of G and H (G/H) were also analyzed based on field observation data. The intra-day distributions of G/Rn and G/H at each site are shown in **Fig. 2**. During data processing, data points with absolute values greater than 10 in the G/Rn or G/H daily series of each period were deleted. Outliers in the G/Rn or G/H series were then removed using the outlier detection method. The G/Rn varied significantly in different half-hour periods of the intra-day, and among the different sites (**Fig. 2-a**). The variation in G/Rn among the sites was lower during the daytime than that at night. The variation range of G/Rn among all sites was generally approximately 0.2 in the daytime. This indicates that the G/Rn of all sites showed high consistency during these periods. At night, the variation mostly ranged between 0.6 and 0.8. In other words, G/Rn was more consistent across sites during the daytime than that at night. The slope and $R^2$ of the linear fitting curve were -0.012 and 0.92, respectively. The $R^2$ of the polynomial fitting curve reached 0.98 (**Fig. 2-c**).

G/H also varied significantly in different half-hour periods of the intra-day and among the different sites (**Fig. 2-b**). The variation in G/H during each period was greater than that of G/Rn. In other words, G/H was less consistent across sites than G/Rn. Like G/Rn, the variation of G/H among sites during the daytime was significantly lower than that at night. The variation range of G/H among all sites in the daytime was approximately 1.0, while the corresponding range at night was approximately 2.0. In most half-hour periods, the mean and median values of G/H at all sites showed significant differences, with the former generally being approximately 0.5 higher than the latter. During 6:30-16:30, the median and mean values generally showed a unimodal distribution, and the $R^2$ of the polynomial fitting curve for the mean series was 0.95 (**Fig. 2-d**).

**Figure 2**.

**3.2 Temporal and spatial analysis of the empirical relationship between G and Rn and between G/Rn and NDVI**

The empirical relationship between G, Rn, and H was analyzed based on the measured data. Overall, G had a strong correlation with Rn but a relatively weak correlation with H. **Figure 3** shows the $R^2$ and slope of the linear fitting between G and Rn in each half-hour period of the intra-day. In each period, G and Rn showed a strong linear correlation at most sites, with a fitted $R^2$ generally above 0.4. The mean and median $R^2$ values of all sites were mainly between 0.5 and 0.8. The strong correlation between G and Rn indicates that it is reasonable to use Rn to calculate G in the RS based energy balance ET models. However, the correlation between G and Rn varied during the different periods. The correlation is relatively high in the periods around 6:00 and 18:00. Especially around 6:00, the $R^2$ of the linear fitting between G and Rn was greater than 0.7 at most sites, and the median $R^2$ of all sites reached 0.8. The correlation between G and Rn in the night periods (20:00-4:30) was slightly stronger than that in the daytime periods (8:00-16:00). During the night periods, the $R^2$ of most sites was generally between 0.45 and 0.70, and the mean and median $R^2$ of all sites were mainly between 0.55 and 0.60. The $R^2$ of most sites generally ranged from 0.40 to 0.65 in the daytime periods, and the mean and median $R^2$ of all sites were concentrated around 0.50. The correlation between G and Rn was relatively low in the period from 10:00 to 15:00, and $R^2$ was lower than 0.65 at most sites, especially in the periods around 14:00, with the mean and median $R^2$ below 0.50.

The slope of the linear fitting of G and Rn in each half-hour intra-day period is shown in **Fig. 3-b**. The fitting slope showed significant differences among the different sites, ranging from 0.1 to 1.1. However, the slopes fitted at most sites also exhibited certain characteristics of a centralized distribution. In each period, the range of the slope at most sites was within 0.3, especially during the daytime periods (8:00-17:30), when the range of the slope was within 0.15. The slopes of most sites were relatively stable during all periods except for the periods of 6:00-7:00 and 17:00-18:00.

In terms of seven land cover types, the intra-day performance of each land type was similar to that of all sites except the Other type (**Fig. 3-c** and **3-d**). The correlation between G and Rn was relatively high during 6:00-7:00 and 17:00-18:00. The correlation in Other and Wetland types is generally higher than that of other land cover types. In each period, the median $R^2$ of all sites in the two types generally exceeded 0.60, and the highest value even exceeded 0.80. Except Other type, the difference of correlation between G and Rn in different land types is mainly reflected in the daytime period except Other type. The correlation in the Forest and Savanna types was significantly lower than that of other types during daytime, especially for Savanna sites, most of which had $R^2$ lower than 0.5 during daytime. In Other type sites, the correlation between G and Rn in the daytime is stronger than that in the night periods. The slope value of each land cover type in the daytime is lower than that in the night. This intra-day distribution of slope was consistent with that of all sites.

**Figure 3.**

The empirical relationship between G and Rn not only varied significantly at different intra-day periods, but also showed great spatial differences among the different sites. The linear fitted $R^2$ between the daily series of G and Rn at each site is

shown in **Fig. 4**. As the median $R^2$ of 48 half-hour periods at each site (**Fig. 4-a**), 91% of the sites (210/230) showed an $R^2 > 0.4$. The linear fitting $R^2$ between G and Rn was >0.6 for 49% of the sites (114/230). The mean $R^2$ for all sites was 0.58. This indicated that the G in most half-hour periods had a strong correlation with Rn at most observed sites. However, there were also 20 sites where the $R^2$ ranged from 0.2 to 0.4. These sites were mainly located in the middle- and low-latitude regions, but were distributed across all observed continents. The correlation was generally stronger in the Northern Hemisphere than that in the Southern Hemisphere, with the mean $R^2$ of the northern sites significantly higher than that of the southern sites. There was a strong correlation between G and Rn at most Eurasian, North American, and Oceanian sites, with a linear fitted $R^2$ generally exceeding 0.4. It showed a relatively weak correlation at many African and South American sites, with an $R^2$ value of less than 0.4. At different latitudes, the strongest correlation between G and Rn was found at the middle and high latitude (>45 °) sites, with the highest $R^2$ values. The $R^2$ values of these sites exceeded 0.4, with a mean value of 0.65. It showed a relatively weak correlation at tropical (<23.4 °) sites. The $R^2$ values of these sites were relatively low, with a mean value of 0.48.

The spatial distribution of the linear fitting $R^2$ of G and Rn at 10:30 and 13:30 (**Fig. 4-b** and **4-c**) was consistent with that in **Fig. 4-a**, while the $R^2$ value was generally lower than the median of each half-hour period. At 10:30, the mean $R^2$ of all sites was 0.52. 26% of the sites (60/230) had $R^2$ values lower than 0.4, and 11 sites had $R^2$ values lower than 0.2. Although mainly distributed at low latitudes (<30 °), these sites were found on all the observed continents. The highest $R^2$ values were at high-latitude (>60 °) sites, with an average of 0.70. The $R^2$ values of low-latitude sites were significantly lower than those of other sites, with a value less than 0.4 for most sites, and an average of only 0.33. The correlation between G and Rn at 13:30 was weaker than that at 10:30, with $R^2$ values slightly lower than those at 10:30. There was a relatively weak correlation between G and Rn at all African sites ($R^2 < 0.4$). Overall, the results showed a strong correlation between G and Rn at most observed sites. It is reasonable to simulate a daily series of G values from Rn in most areas. However, it is necessary to apply this relationship cautiously in some areas at mid-low latitudes, especially in tropical areas.

For different land cover types, the correlation between G and Rn was the strongest at Other and Wetland sites. The mean value of the median $R^2$ of statistical sites was 0.71 and 0.67 for these two types, respectively. There was also a strong correlation between G and Rn at Cropland, Shrubland and Grassland sites. The mean value of corresponding $R^2$ is about 0.60. There was only one site with a median R2 lower than 0.4 in each of the three land cover types. The mean value of corresponding $R^2$ was 0.55 for Forest sites. However, the correlation was relatively weak in the Savanna type sites, and the mean value of corresponding R2 is 0.49. The weak correlation between G and Rn (R2<0.4) were mainly distributed in the Forest (6/97), Grassland (7/42) and Savanna (3/15) sites.

**Figure 4.**

The empirical correlation between G/Rn and NDVI was also analyzed. The results showed that the correlation between G/Rn and NDVI was weak in the daily series but strong in the monthly series. **Figure 4-d** and **4-e** show the linear and exponential

fitted $R^2$ values between the monthly series of G/Rn and NDVI at each observed site, respectively. During data processing, only monthly values of observation days greater than 15 days were used, and monthly values affected by frozen soil or snow cover (mean air temperature below -5 ℃) were excluded. In general, there was a strong correlation between the monthly G/Rn series and NDVI. The fitted $R^2$ values of the linear and exponential functions were consistent. The median and mean $R^2$ of all sites were 0.71 and 0.65, respectively, and 72% of the sites (157/218) had an $R^2$ above 0.60. The exponential correlation between G/Rn and NDVI was stronger than the linear correlation at several sites, including US-Gle and US-Twt. The exponential fitted $R^2$ of the two sites could be increased from 0.55 and 0.31 of the linear $R^2$ to 0.77 and 0.50, respectively. Conversely, the linear correlation was stronger than the exponential correlation at other sites, such as ES-AMO and CN-QIA. The linear fitted $R^2$ of the two sites could be increased from 0.21 and 0.68 of the exponential $R^2$ to 0.43 and 0.83, respectively. Overall, the spatial distribution of the linear or exponential fitted $R^2$ of G/Rn and NDVI was similar to the linear fitted results of G and Rn. The correlation between G/Rn and NDVI was stronger in the Northern Hemisphere than that in the Southern Hemisphere. The mean $R^2$ value of the northern sites (0.69) was higher than that of the southern sites (0.38). It showed the strongest correlation at the middle- and high-latitude (>50 ̊) sites. The $R^2$ values for these sites were generally higher than 0.6, with an average of 0.76. A relatively weak correlation between G/Rn and NDVI was found at the low-latitude sites, with a mean $R^2$ of 0.38. There were 13 and 15 sites showing weak linear and exponential correlations between G/Rn and NDVI, respectively, with a fitted $R^2$ lower than 0.2. These sites were mainly located in Australia and Southeast Asia.

### 3.3 Temporal and spatial accuracy of five G simulation methods

In this study, four criteria were tried to evaluate the model. The results showed that only NSE was suitable for the evaluation of different sites and time periods, whereas RE and KGE were not suitable for the evaluation of different sites, and the RMSE was unsuitable for the evaluation of different time periods. The RE, RMSE, and KGE simulated by the LC method at each site and time period are shown in **Fig. 5**. The RE values tended to be affected by the mean value of the referenced G in **Eq. (7)**. The RE values of all sites were within ±3% (**Fig. 5-a**) during the daytime periods from 7:00 to 16:00 because of the relatively large mean value of G during these periods (**Fig. 1-a**). However, when the mean value of the referenced G was low during the period from 17:00 to 6:30, the RE value at each site was generally greater than that during the daytime periods. In particular, if the mean value of the referenced G is much lower (e.g., close to 0) in a half-hour period for a site, even a small simulation bias could result in an extremely large RE value; for example, the RE of some sites exceeded 8,000% at 5:00 and 5:30 (**Fig. 5-a**). The mean RE value of all sites was also too high (e.g., 300%) during these periods. Therefore, RE is unsuitable for evaluating different time periods or sites. However, the median RE values of all sites may be more robustly used for evaluation than the mean values. Because the RE is included in the formula of the KGE, the shortcomings of the RE are introduced into the KGE. Therefore, KGE is also unsuitable for the evaluation of different time periods or sites. According to **Fig. 5-b** and **Fig. 1-a**, there was a positive relationship between the RMSE and G values during half-hour

periods in the intra-day period. The RMSE values were directly affected by the G values. Due to the significant variations in G values in each period, the RMSE was not suitable for comparison evaluation of simulation accuracy between different periods. The RE, RMSE and KGE simulated by the LC method in each land cover type were shown in **Fig. S1**.


## Figure 5.

The NSE was used to evaluate the accuracy of the five G simulation methods at different sites and at half-hour intervals. Daily series were randomly assigned to one of two datasets: 80% were assigned to the calibration dataset and 20% to the

validation dataset. The process of random assignment was repeated to generate 100 independent datasets. Results showed that there was little difference between the performance of the models in calibration and validation datasets. It indicated that these methods are robust. The performance of the LC_NDVI_P in calibration and validation datasets at some sites could be found in **Fig. S2**. **Figure 6** shows the NSE values simulated by the LC, LC_NDVI_P, LC_NDVI_E, LC_fc_SE, and LC_fc_ST methods. In general, the performance of each method varies significantly among different sites and time periods.

The simulation accuracy of each method showed high consistency among the different half-hour intervals within the intra-day period. It was highest in the period of 6:00-7:00, followed by the period of 17:00-18:00, whereas it was lowest in 10:00-15:30. Regarding the different methods, the accuracy of the first three methods (LC, LC_NDVI_P, and LC_NDVI_E) was significantly higher than that of the last two methods (LC_fc_SE and LC_fc_ST).

The LC method generally demonstrated its ability to accurately simulate the daily series of G at the site scale in each half-

hour period, with the NSE of most sites exceeding 0.40. The mean and median NSE values of all sites were generally higher than 0.50 in each time period. It showed the highest accuracy at 6:00 and 6:30, with the NSE of most sites above 0.70. The accuracy was the lowest from 11:30 to 15:00, with the mean and median NSE values of all stations being between 0.45 and 0.50. The mean and median NSE values of all the sites in the other periods were generally greater than 0.50. This indicates that the simple LC method was able to accurately simulate a half-hourly series of G from Rn at most sites, but also lost its

ability with unsatisfactory accuracy at a few sites.

Although NVDI was embedded in the LC method, the performances of the LC_NDVI_P and LC_NDVI_E methods were similar to those of the LC method. In other words, the consideration of NDVI resulted in limited improvement in the accuracy of the LC method. In addition, the accuracy of the LC_NDVI_E method was significantly lower than that of the first two methods at 14:30 and 15:00, with mean NSE values of only 0.28 and 0.33, respectively. This was because of the

low accuracy (NSE<0.2) of the LC_NDVI_E method at more sites during these two periods.

The accuracy of the LC_fc_SE and LC_fc_ST methods based on fractional vegetation coverage was relatively low (**Fig. 6-d and 6-e**). The two methods showed little difference across sites and periods. The two methods were able to accurately simulate G only at 5:30-7:00 and 17:30-18:00, with the median NSE values of all sites exceeding 0.5. Conversely, the performance was poor in most night and daytime periods, such as 20:00-4:30 and 8:30-15:30. The NSE of most sites was

below 0.4, and the mean and median NSE values of all sites were below 0.2. This indicates that the application of these two methods considering fractional vegetation coverage requires caution in the G simulation.

**Figure 6.**

**Figure 7** shows the NSE simulated by each method in seven land cover types. The intra-day performance of each land cover type was similar to that of all sites except for the Other type, with the highest simulation accuracy at the periods of 6:00-7:00 and 17:00-18:00. The intra-day accuracy varied greatest at the Forest and Savanna sites. The median NSE of all sites simulated by the LC_NDVI_E method was close to 0.8 at the period of 6:00-7:00, while the corresponding NSE was only approximately 0.4. It varied little at other land cover types, especially for Wetland and Shrubland types. The greatest and

lowest values of median NSE for all sites simulated by the LC_NDVI_E method were approximately 0.7 and 0.6, respectively. The NSE of the LC, LC_NDVI_P and LC_NDVI_E methods showed a unimodal distribution in the Other type sites. The NSE was significantly higher in the daytime than at night periods. The highest value was in the morning and noon periods, with the median NSE of all sites exceeding 0.8. The model performance was significantly better than other land cover types. In the Other type sites, the LC_NDVI_E method performed better than other methods, with the median NSE

higher than 0.6 in each time period.

**Figure 7.**

The spatial distribution of the NSE simulated by the LC method at each site is shown in **Fig. 8**. Overall, there were

significant differences in the performance of the LC method among sites, with the lowest and highest NSE values of each site being -0.37 and 0.94, respectively. As for the median NSE of 48 half-hour periods at each site (**Fig. 8-a**), the performance of the LC method was satisfactory at most sites, with the mean NSE of all sites being 0.58. The NSE values of 70% of the sites (160/230) were higher than 0.5. However, 27 and 5 sites had NSE values lower than 0.4 and 0.2, respectively, indicating that the performance was poor at these sites. For different latitudes, the performance was generally

satisfactory at the middle and high latitudes, with NSE values above 0.4. The best performance was observed at high latitudes, with a mean NSE value of 0.69. The accuracy of the LC method was generally low at tropical sites, with a mean NSE of 0.47. The performance was generally satisfactory at most sites in Eurasia and North America, with NSE values higher than 0.5. The NSE values of many sites exceeded 0.7 in these regions. Conversely, relatively poor performance was found in the African, South American, and Oceanian sites, especially in African sites.

The LC method accurately simulated G at 6:30 in most sites (**Fig. 8-b**), with the mean and median NSE values of all sites being 0.73 and 0.78, respectively. The sites with the NSE higher than 0.5 and 0.6 took up 92% (211/230) and 84% (193/230), respectively. The NSE was higher than 0.5 at all sites of Eurasia and North America, and the NSE of most sites exceeded 0.7.

The method was also able to accurately simulate G at 18:00 in most sites (**Fig. 8-e**), with a mean NSE of all sites being 0.62. 82% (188/230) and 59% (135/230) of sites had NSE values exceeding 0.5 and 0.6, respectively.

The LC method performed poorly at many sites at 10:30 (**Fig. 8-c**) and 13:30 (**Fig. 8-d**), where the mean NSE values of all sites were 0.49 and 0.47, respectively. The sites with an NSE exceeding 0.5 only accounted for 51% (118/230) and 43% (100/230) during the two half-hour periods, respectively. The method still performed well at high-latitude sites, with a mean NSE of 0.69 and 0.65, respectively. Conversely, it lost the ability to accurately simulate G at tropical sites. The NSE values at most sites were lower than 0.5, with a mean NSE of only 0.29. The performance of the method was poor at the African

and South American sites, with NSE values of each site below 0.5.

For different land cover types, the LC method performed better in the Cropland, Wetland and Other type sites. The mean value of median NSE of Wetland and Other sites was 0.66 and 0.69, respectively. The method was also able to accurately simulate G in the Forest, Grassland and Shrubland type sites, with the corresponding mean NSE of 0.57 or 0.56. It performed the worst at the Savanna sites, with the corresponding mean NSE was only 0.47. Since the Savanna sites are mainly

distributed in tropical regions, this is consistent with the relatively poor performance of tropical region site as mentioned above. The performance of the method varied significantly in each land cover types except for the Other type sites. In the Wetland type sites, there were 3 sites in the United States with the NSE value lower than 0.3. The NSE of other 35 sites was higher than 0.50, with the highest value was close to 0.90. The Grassland sites were distributed in Asia, Europe, North America and Oceania. The NSE value was greater than 0.5 at each Grassland site in Europe. Cropland sites were distributed

in Asia, Europe, and the United States. The NSE value was lower than 0.60 at 8 sites in the United States, with the mean NSE value of only 0.45. The method was able to accurately simulate G at 11 sites in Europe except for one site in Mediterranean region, with the mean NSE value of 0.74. The NSE for the two Asian sites was 0.54 and 0.71, respectively.

**Figure 8.**


According to the evaluation of the five methods mentioned above, LC, LC_NDVI_P, and LC_NDVI_E performed well. **Figure 9** shows the optimal values of the parameters of the three methods at each site and half-hour periods. As for the results of the LC method (**Fig. 9-a**), on the one hand, the optimal parameter values varied significantly in different sites and half-hour periods. The median optimal parameter of all sites was generally around 0.4 in daytime periods and 0.7 at night.

The optimal parameter of each site ranged from 0.4 to 1.1 in the night periods. In contrast, the optimal parameters of most sites showed a concentrated distribution. For example, the optimal parameter values of most sites were generally concentrated in the range of 0.25-0.45 during 8:30-16:30, while the corresponding optimal values were generally concentrated in the range of 0.6-0.85 during 19:30-5:30. The median optimal value of all sites was stable at about 0.35 and 0.75 during these two time periods, respectively. The optimal parameter values of the LC_NDVI_P method showed little

difference from those of the LC method at each site and half-hour period (**Fig. 9-b**). The optimal parameter (b in **Eq. (4)**) values of the LC_NDVI_E method ranged from 0.01 to 1.4 among different sites and half-hour periods (**Fig. 9-c**). Similar to

the LC and LC_NDVI_P methods, the optimal parameter values of the LC_NDVI_E method also showed a concentrated distribution in each period, especially during 7:30-15:00 and 19:30-5:00. Almost all sites with the optimal parameter values ranged from 0.01 to 0.4, and the optimal values of most sites were concentrated in the range 0.06 to 0.25 during these two
periods.

**Figure 9.**

## 4 Discussion

### 4.1 Limitations and uncertainties

Theoretically, surface energy is balanced. The energy unclosure might be mainly caused by the error of the observed data. The observed G' instead of G was generally used to investigate the energy balance ratio (Wilson et al., 2002). The energy balance closure problem might be largely caused by the soil heat storage (Foken, 2008). Compared with G, other energy terms can be observed more accurately. The eddy covariance measurements of H and LE are generally considered to be the most accurate observations available, and have been widely used as a reference to evaluate other simulation methods. The
**Eq. (1)** makes full use of the surface energy term that can be accurately measured at present. In other words, it assumes that the measurements of Rn, H and LE are accurate in this study. The uncertainties of measurements are not considered in this study. However, the uncertainty of G estimated by the SEBR method could be very large at some sites, which have low magnitude of G. Although the majority of sites have G values greater than 10 W/m$^2$, and take up more than 15% of Rn, there are some sites (20/230) with G values lower than 5 W/m$^2$. There would be great uncertainties of the SEBR method in
simulating G values at these sites.

The accuracies of the LC_fc_SE and LC_fc_ST methods, which embed fractional vegetation coverage in the G simulation, were satisfied for monthly simulation, but were significantly lower than those of the other three methods in simulating daily values. The weak correlation between G/Rn and NDVI might be the main reason for the poor performance of these methods. However, the coarse-resolution NDVI data used in this study are not sufficient to represent the scale of flux measurements,
especially for sites with heterogeneous land surface. This might be the main reason for this weak correlation. The application of higher resolution and continuous vegetation index data series is expected to improve the simulation accuracy of these methods. A large error in the G simulation might be induced in the ET modelling process, thereby reducing the accuracy of the ET estimates. In RS ET models, Rn is generally calculated using radiation balance with RS images and meteorological inputs. However, observed Rn was used for simulating G in this study. In other words, it was assumed that Rn is accurately
simulated by the RS ET models. Therefore, it should be noted that the uncertainty in Rn calculation was also a source of error in G simulations in ET models.

The evaluation results of this study are expected to provide reference for RS ET model application and developers. For example, the performance of these methods was good and poor at some sites and time periods and some land-cover types.

RS ET modelers could check the advantage of the models at good performance regions, and find why the models are poor at some other areas, then revise the models to improve the accuracy at poor performance regions. However, how to improve the model to improve the accuracy needs further research.

## 4.2 Temporal and spatial variations in the simulation of G

Intra-day and daytime G accounted for 19.2% and 28.8% of Rn, respectively. This indicates that G plays an important role and cannot be ignored in RS ET models. Ignoring G would lead to a great deviation in ET estimates, including the term (Rn-G). The intra-day and daytime G' values were only 4.7% and 24.1% of the corresponding G, respectively. This indicates that the measured G' must be carefully used in the application of the surface energy balance equation. If G' was used instead of G in the equation, the actual G would be largely underestimated. This is consistent with the results of Meyers and Hollinger (2004) and Lu et al. (2018).

Several methods for simulating G from Rn in energy balance-based RS ET applications have been evaluated in this study. G/Rn, which varied intra-day, was the key to these methods. Santanello and Friedl (2003) provided diurnal covariations in G and Rn. In this study, more concise linear and polynomial functions of the G/Rn daytime distribution were fitted (**Fig. 2**) from 230 observation sites worldwide. The linear- and polynomial-fitted $R^2$ values were 0.92 and 0.98 respectively.

The evaluation results show that the accuracy of the G simulation varied significantly in different half-hour intra-day periods. The highest accuracy was exhibited during 6:00-7:00 and 17:00-18:00. It also showed a slightly higher accuracy during night periods than during daytime periods. The lowest accuracy is observed at noon. This is consistent with the correlation between G and Rn, indicating that the accuracy of the G simulation is affected by the correlation between Rn and G. In other words, the stronger the correlation between Rn and G, the higher the accuracy of the simulation of G from Rn. However, RS ET models are generally applied during daytime periods. For example, MODIS data represents conditions around 10:30 and 13:30, but the simulation accuracy of these two periods is the lowest during intra-day periods. This is an urgent issue to be solved for G simulation in RS ET applications, which requires the attention of RS ET modelers.

The performance of G simulation methods also showed significant spatial variation. The accuracy of the G simulation varied significantly among the observation sites, with the corresponding NSE ranging from 0.2 to 0.9. The G simulation of most sites showed high accuracy in most half-hour periods. This verified the reliability of global RS ET products in these regions because the more accurate G simulation provided a guarantee for an accurate ET simulation, such as in Eurasia and North America. However, there were also some sites with low simulation accuracy, such as most sites in Africa, South America, and Oceania. A large error in the G simulation would be induced in the ET simulation results and reduce the reliability of ET estimates. Therefore, the application of RS ET estimates and products in these areas needs more caution for its accuracy. The spatial distribution of the model accuracy was also consistent with the correlation between G and Rn. The sites with satisfied model performance were generally characterized by seasonal variation in G and Rn due to the climate in these regions. Conversely, the sites with poor model performance showed little seasonality. Whether G and Rn have seasonal variations was also an important factor affecting the accuracy of empirical methods in simulating G. This was consistent with the

evaluation results of the LE simulation accuracy (Liu, 2021; 2022). This study analysed the performance of the five methods in seven land cover types. The methods performed better in the Wetland and Other type sites than other ones. RS ET modellers were recommended to take more cautions in G simulation at the Savanna type sites, because the simulation accuracy was generally lowest at this type sites. In addition to climate and land cover factors, regional soil and other environmental factors might also be important factors affecting the accuracy of G simulation. More evaluations of G simulation at regional scale are recommended for further research.

### 4.3 Applicability of common G simulation methods in RS ET models

Daytime RS images are generally applied in ET models. The evaluation results of 230 worldwide observation sites showed that the optimal parameter values of most sites were generally concentrated in the range of 0.25-0.45 during daytime periods, with the median of all sites being stable at approximately 0.34. This indicates that the coefficient values applied in most ET models were reasonable, but the coefficient values applied in the GLEAM model were relatively low.

In the RS ET models, fixed empirical parameters were applied to the global terrestrial G simulation. The fixed parameters might be suitable for some regions, but not on a global scale. This study confirmed that the optimal parameter values vary significantly from site to site. Fixed parameter values induced large errors in the G simulations in other regions. Therefore, it is recommended that model developers consider the spatial variations of G simulation parameters in RS ET modeling on a global scale.

Some RS ET models embed vegetation indices (e.g., NDVI, LAI, or fractional vegetation coverage) or LST into the coefficient of the LC method, such as the SEBAL and SEBS models. Evaluation of the LC_NDVI_P and LC_NDVI_E methods, which are the LC methods embedded by the NDVI, showed that the improvement in simulation accuracy was limited by considering the NDVI. The mismatch between flux observations at the site scale and vegetation index data at the grid scale may be one of the reasons for this result. In addition, the term containing the NDVI in these methods could be taken as a whole, which is similar to the coefficient in the LC method. Therefore, the performance of this method is expected to differ slightly from that of the LC method when the parameter is optimally calibrated. Other models embedded by the LAI (Choudhury, et al., 1987; Allen et al., 2011) and LST (Bastiaanssen, 1995; Faridatul, 2020), which were not evaluated in this study due to data limitations, may have similar performances, such as the METRIC model embedded by the LAI and the modified SEBAL model (Faridatul et al., 2020) embedded by the LST and NDVI. Saadi et al. (2018) evaluated three such methods using data from a single observation site. Their results showed that the accuracy order was the Bastiaanssen (1995) method > Jackson et al. (1987) method > Choudhury et al. (1987) method, with RMSE values of 0.09, 0.15, and 0.2, respectively. Evaluation of such methods embedded with the LST data is recommended for further research where data is available. The results of this study indicate that the performance of the different methods varied at some sites. However, the differences among the methods were not significant at global sites as a whole.

In general, the LC method and the methods embedded with the NDVI accurately simulated half-hourly G series at most global sites. There was little difference in simulation accuracy between the different models. However, the performance was

505 poor at some sites. Moreover, the optimal values of the model parameters differed among the different sites. This has also verified by Chen et al. (2019). These issues need to be considered in RS ET models to improve simulation accuracy. The optimal parameter values for each site showed relative stability between different half-hour periods in the daytime, indicating that it was feasible to apply the same coefficient value in different daytime periods.

## 5 Conclusions

Intra-day and daytime G accounted for 19.2% and 28.8% of Rn, respectively. This indicates that G plays an important role and cannot be ignored in RS ET models. The accuracy of the five G simulation methods in energy balance-based RS ET models was evaluated using half-hourly observations from 230 flux sites. The LC method and the methods embedded with the NDVI could accurately simulate a half-hourly G series at most sites. The mean and median NSE values of all sites were generally higher than 0.50 in each half-hour period. However, the two methods embedded by fractional vegetation coverage

in the G simulation showed poor performance in most half-hour periods, except for the periods of 6:00-7:00 and 17:00-18:00, with mean and median NSE values of all sites below 0.20. The poor performance might be mainly caused by the coarse-resolution vegetation index data, which could not represent the scale of flux measurements. The performance of each method was generally consistent at different sites and time periods.

The accuracy of each method varied significantly at different sites and at half-hour intervals. The highest accuracy was

520 exhibited during 6:00-7:00, followed by the period of 17:00-18:00. There were 92% (211/230) sites with an NSE of the LC method greater than 0.50 at 6:30. It showed a slightly higher accuracy during night periods than during daytime periods. The lowest accuracy was observed at noon periods (10:00-15:30). For example, the sites with an NSE exceeding 0.50 only accounted for 51% (118/230) and 43% (100/230) at 10:30 and 13:30, respectively. The NSE values of the different sites ranged from -0.37 to 0.94. The accuracy of the Northern Hemisphere sites was generally higher than that of the Southern

Hemisphere sites. In general, it showed the highest accuracy at high-latitude sites, followed by middle-latitude sites, and exhibited the lowest accuracy at low-latitude sites, especially at tropical sites. As for the median NSE of 48 half-hour periods in the LC method, the mean NSE value of the high latitudes and tropical sites was 0.69 and 0.47 respectively. The performance of the LC method and the methods embedded with the NDVI were generally satisfactory at the Eurasian and North American sites, with the NSE values of most sites exceeding 0.70. Conversely, it exhibited relatively poor

performance at the African, South American, and Oceanian sites, especially the African sites. Both the temporal and spatial distributions of the accuracy of the G simulation were positively correlated with the correlation between G and Rn. In other words, the sites or periods with stronger correlations between G and Rn have higher simulation accuracy.

Overall, the LC, LC_NDVI_P, and LC_NDVI_E methods accurately simulated the G series at most observation sites and half-hour periods in the intra-day, with an NSE value exceeding 0.50. However, the performance of these methods was poor

at some sites and time periods. This negatively affects the accuracy of energy balance-based RS ET simulations. The application of RS ET datasets covering these sites requires caution. The performance was best in the Wetland and Other type

sites, and was worst at the Savanna type sites. Improvement of G simulation at low-accuracy regions is recommended for the RS ET modelers, such as low-latitude regions and Savanna type sites. The weak correlation between the G and Rn is the physical reason for the poor accuracy of G simulation in these regions and sites. Instead of the Rn, finding another variable that has a physical connection and strong correlation with the G might be a more efficient solution to improve the accuracy of the empirical estimation method for G. In addition, the optimal parameter value of each method varied significantly at different sites. Therefore, the fixed parameter values in the G simulation methods do not match the actual situation. Variable parameters are recommended in empirical methods of G simulation to improve accuracy.

**Data availability:** The FLUXNET and TERN OzFlux datasets can be downloaded freely from https://fluxnet.org/data/download-data/ and http://www.ozflux.org.au/.

**Author contribution:** Z. F. conducted the analysis and wrote the manuscript.

**Competing interests:** The author declares that he has no conflict of interest.

**Acknowledgments**

This study was supported and funded by the National Natural Science Foundation of China (42171029) and the Strategic Priority Research Program of the Chinese Academy of Sciences (XDA2006020202; XDA23090302). We are grateful to the Data Portal serving the FLUXNET community (https://fluxnet.org/) and OzFlux (http://www.ozflux.org.au/index.html) for providing FLUXNET and TERN OzFlux datasets. We would also like to thank D. Papale, G. Pastorello, and J. Beringer for their kind assistance with data access. We thank the editor B. Su and four anonymous reviewers for their constructive comments, which helped to improve the manuscript.

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

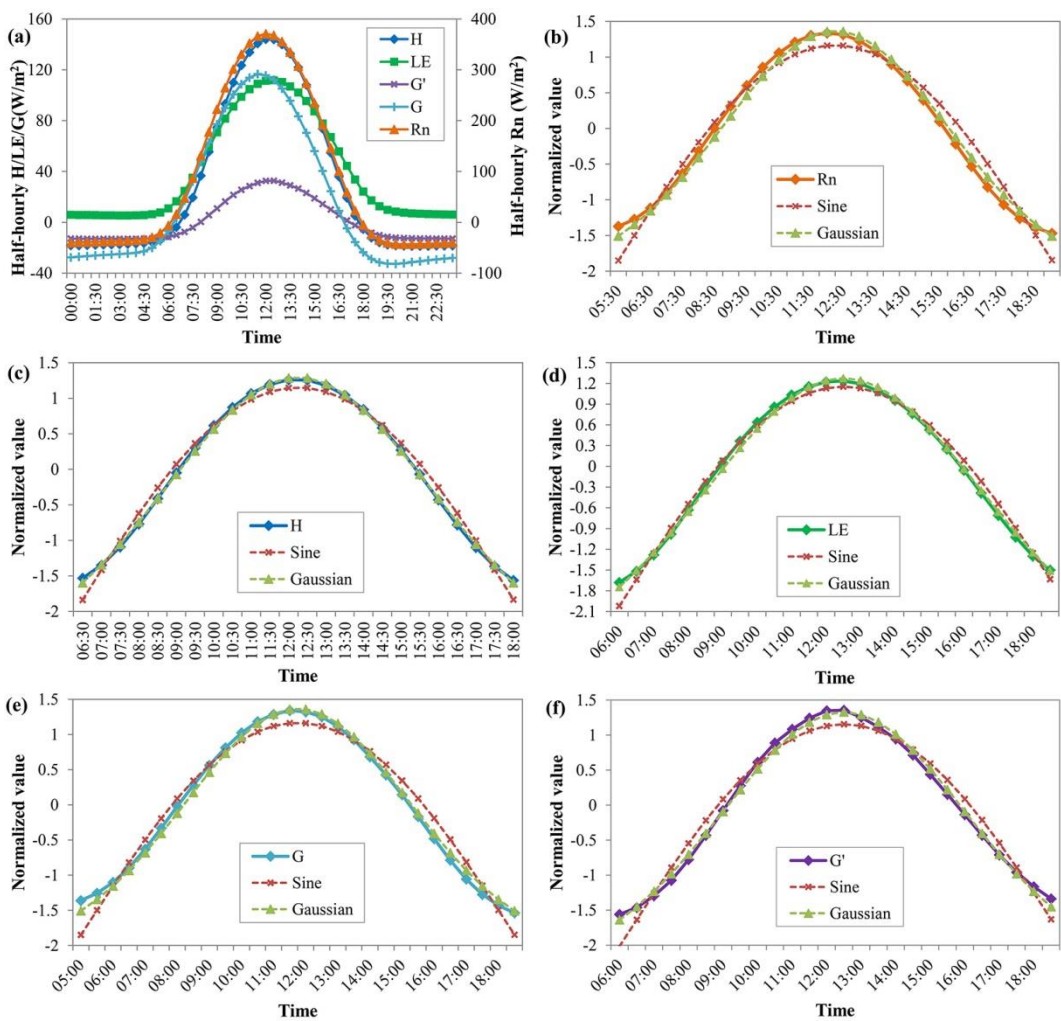

**Figure 1: Intra-day distribution of raw and normalized Rn, H, LE, G, G', and the values from the Sine and Gaussian functions (a is raw Rn, H, LE, G, and G'; b-f are normalized Rn, H, LE, G, and G', respectively).**


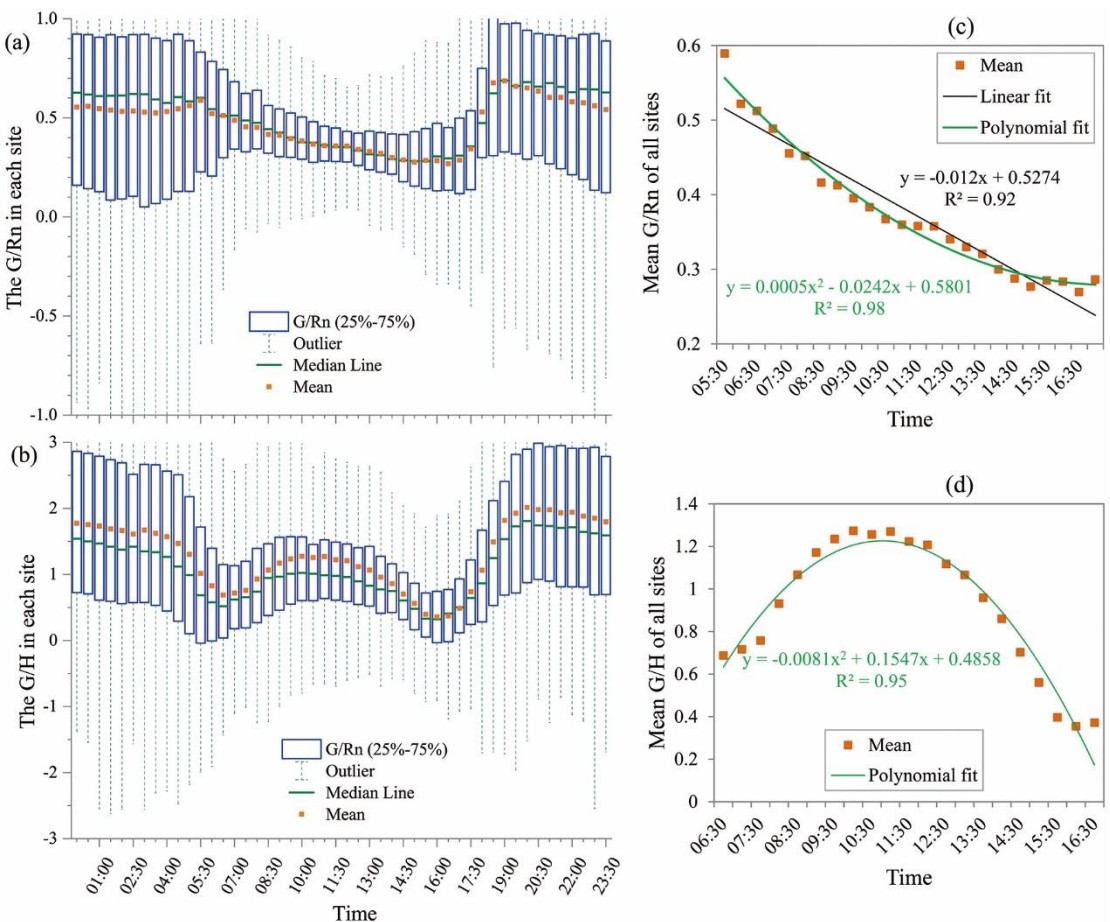

**Figure 2: Intra-day distribution of the ratio of G and Rn (G/Rn) and the ratio of G and H (G/H) in each site (a and b are intra-day distributions of G/Rn and G/H ratios in each site, respectively; c and d are the fitted lines of the mean value of G/Rn and G/H in the daytime, respectively).**

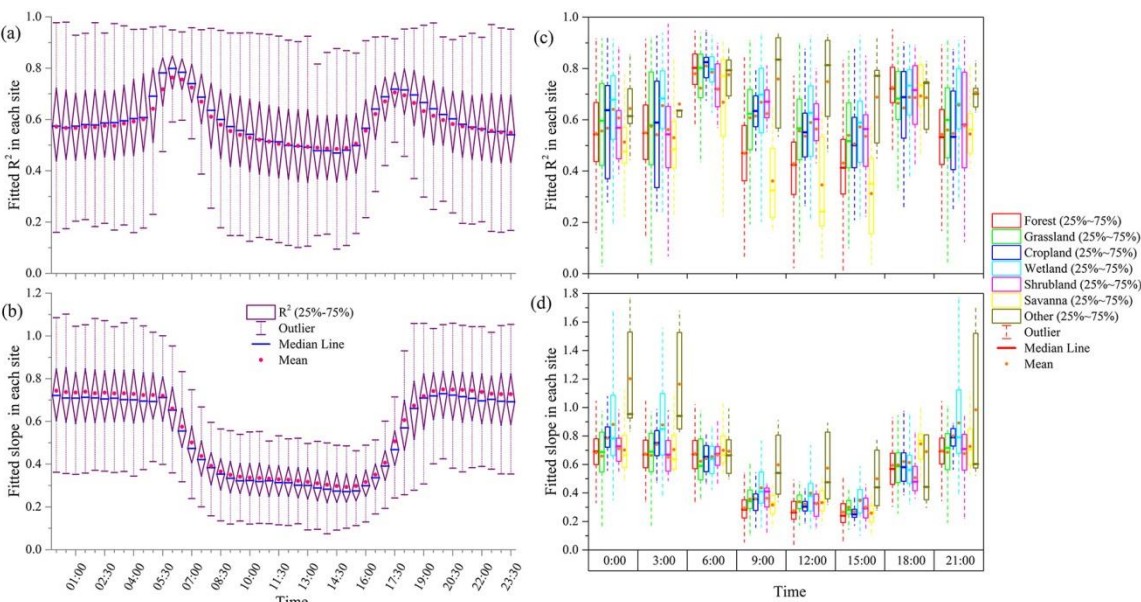

Figure 3: Intra-day distribution of the linear fitted R² (a and c) and slope (b and d) between G and Rn (c and d are classified by land cover types).

**770**

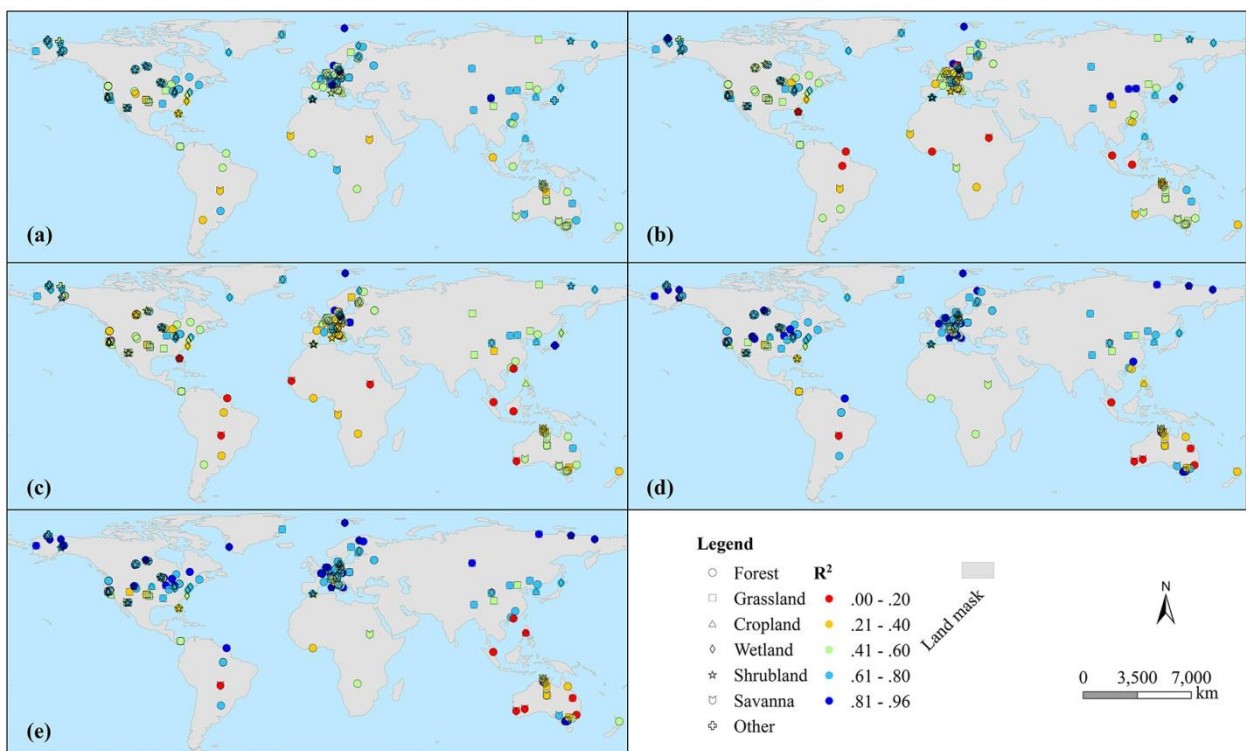

**Figure 4: The linear or exponential fitting $R^2$ between daily series of G and Rn, and that between monthly series of G/Rn and NDVI at each observed site (a is the median $R^2$ of 48 half-hour periods at each site; b and c are the $R^2$ values between G and Rn at 10:30 and 13:30 respectively; d and e are linear and exponential functions fitting $R^2$ between G/Rn and NDVI, respectively).**

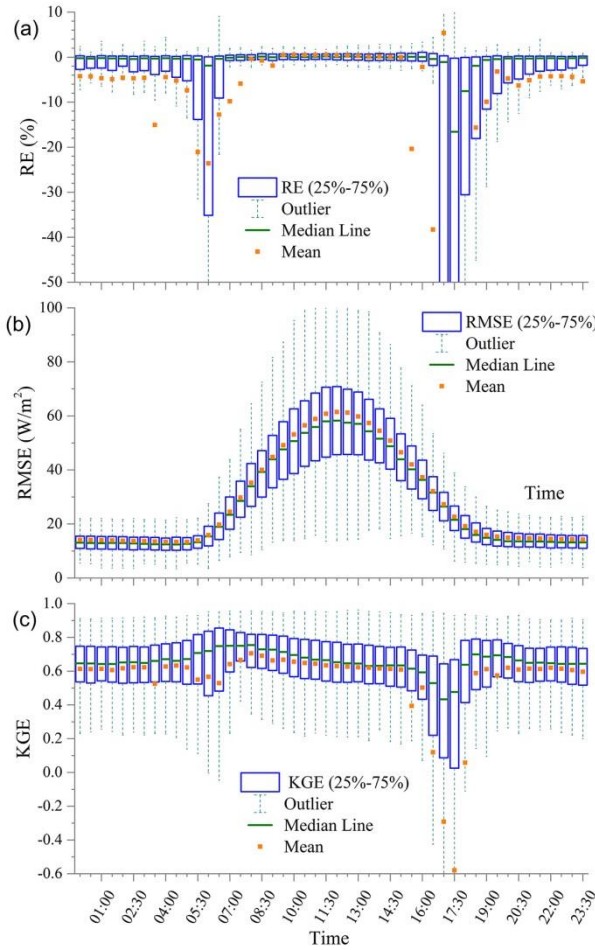

**Figure 5: The RE, RMSE and KGE simulated by the LC method in each site and half-hour intervals.**

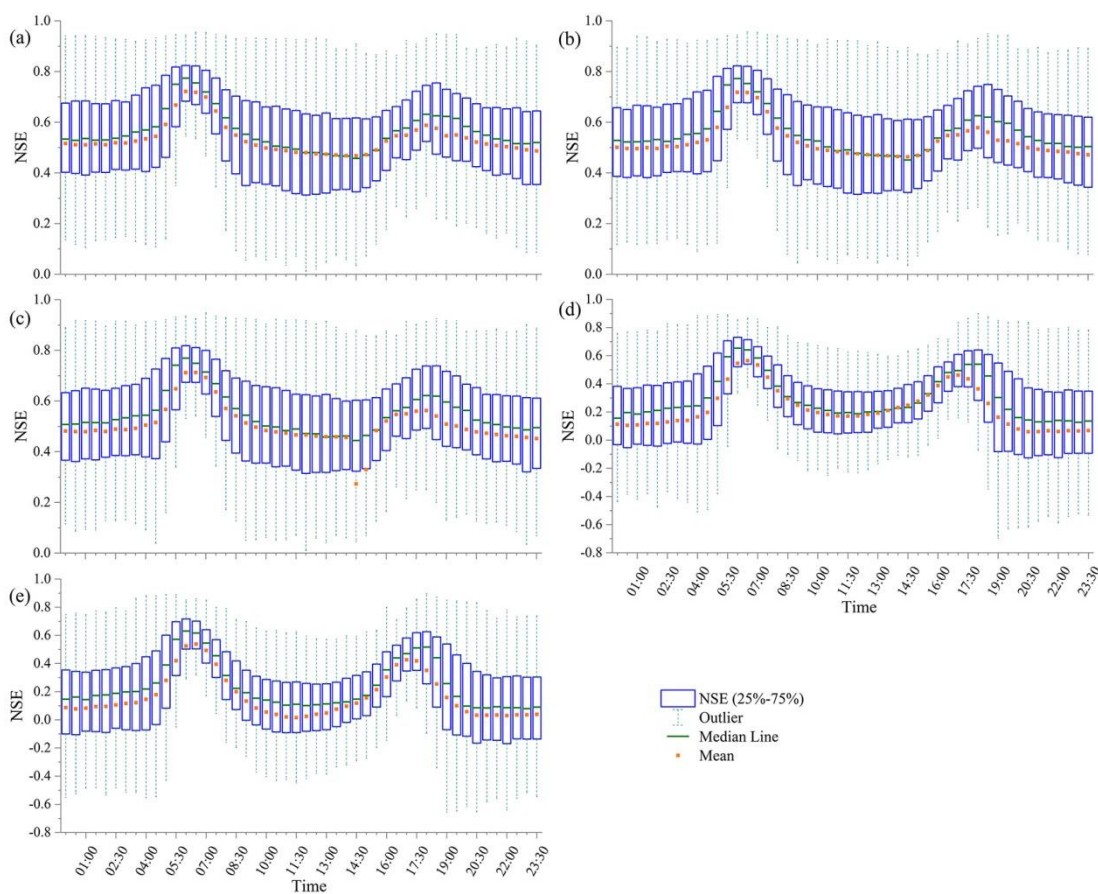

**Figure 6: The NSE simulated by the (a) LC, (b) LC_NDVI_P, (c) LC_NDVI_E, (d) LC_fc_SE and (e) LC_fc_ST methods based on optimized parameters in each site and half-hour intervals.**

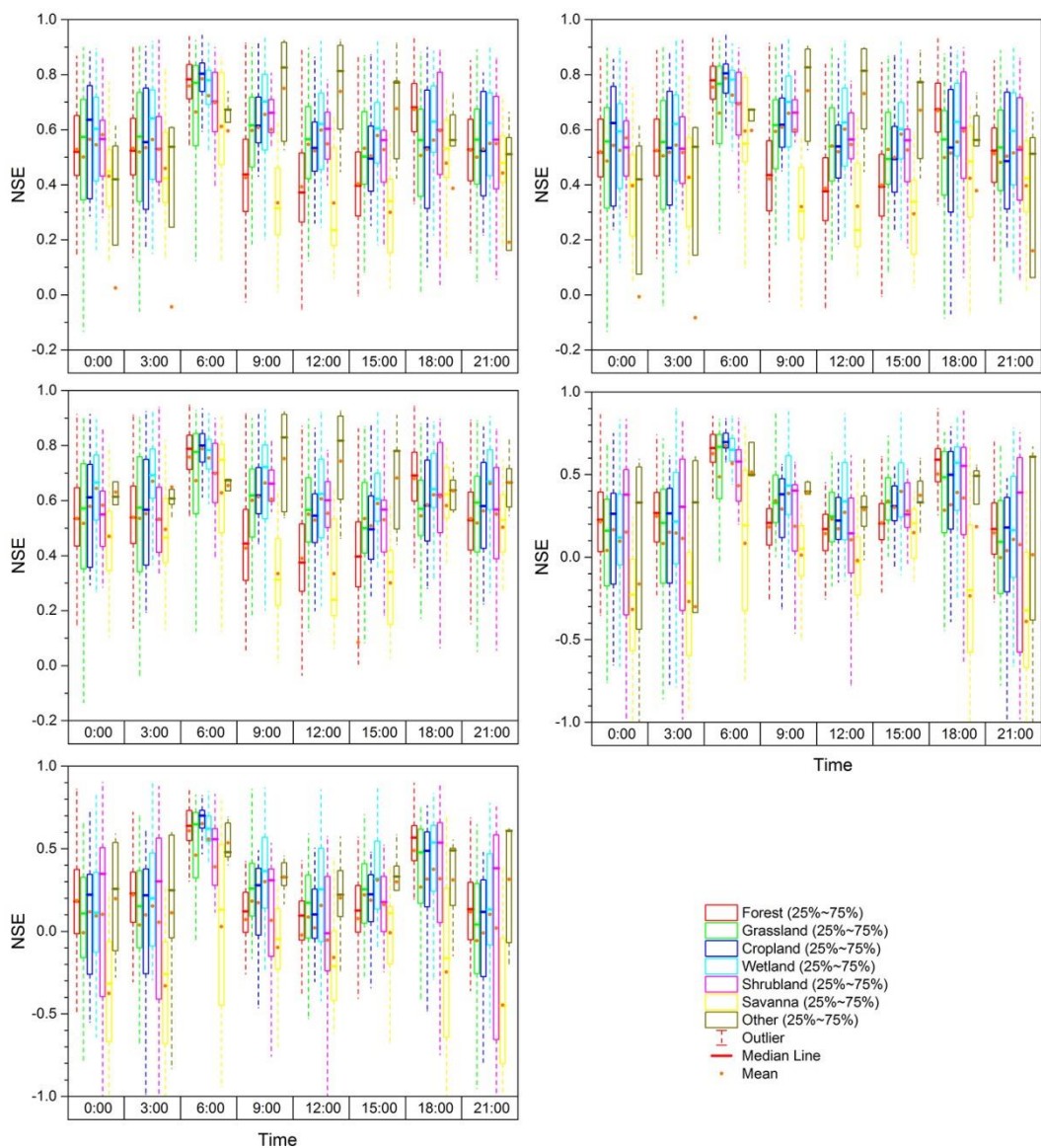

**Figure 7: The NSE simulated by the (a) LC, (b) LC_NDVI_P, (c) LC_NDVI_E, (d) LC_fc_SE and (e) LC_fc_ST methods in each land cover type.**

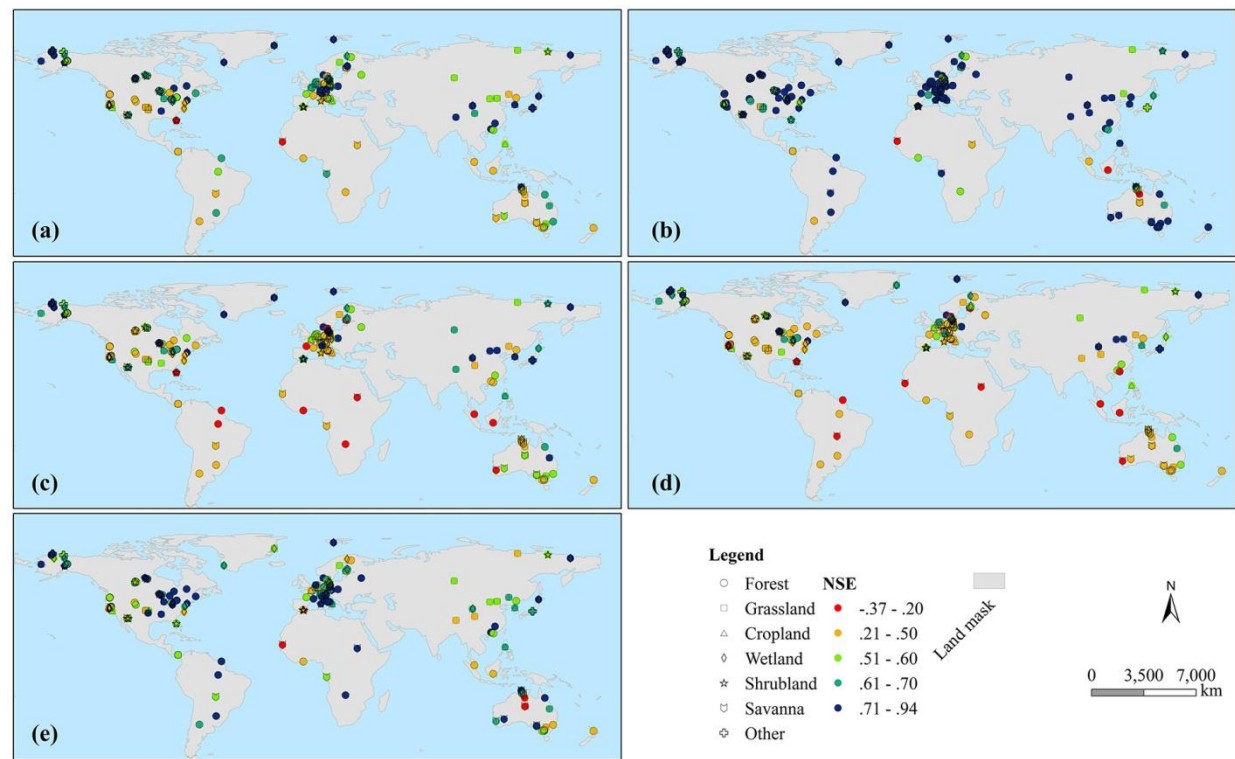

**Figure 8: Spatial distribution of NSE values simulated by the LC method (a is the median NSE of 48 half-hour values; b-e represent the NSE values at 6:30, 10:30, 13:30 and 18:00, respectively).**

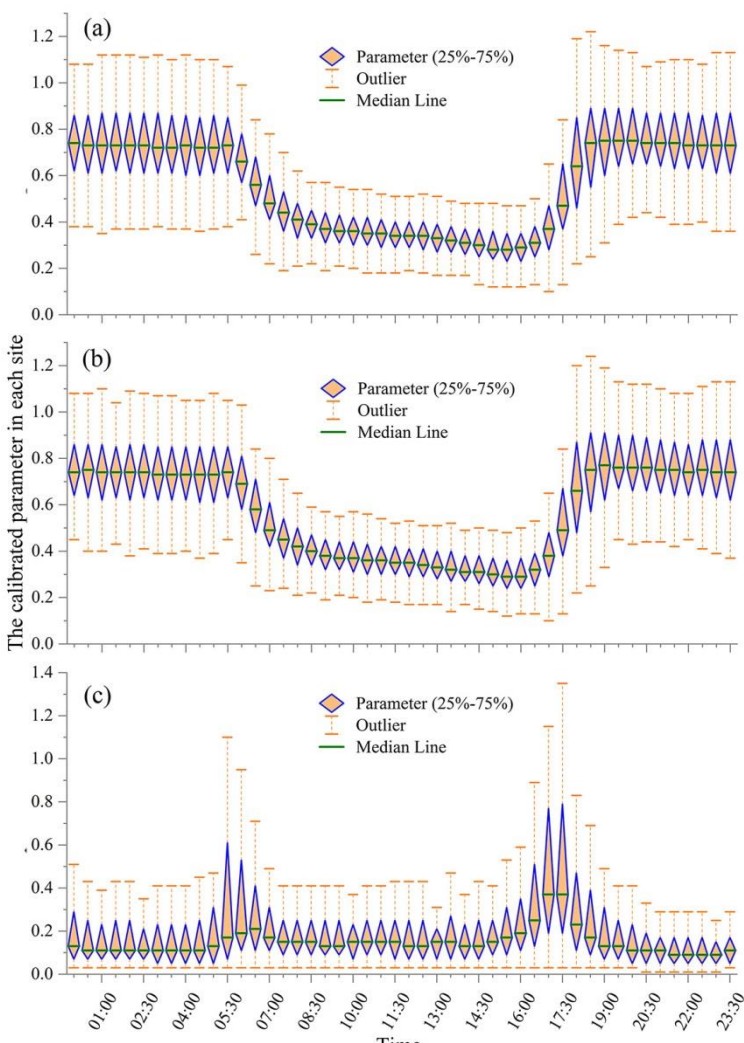

Figure 9: The optimal parameters of the (a) LC, (b) LC_NDVI_P, and (c) LC_NDVI_E methods in each site and half-hour period.