# Peer review of "Accuracy of five ground heat flux empirical simulation methods in the surface energy balance-based remote sensing evapotranspiration models"

_Hydrology and Earth System Sciences, 2022_

## Author Comment (AC1)

**Response (Referee #1 comment)**

**Ms. Ref. No.**: hess-2022-125

**Revised title**: Accuracy of five ground heat flux empirical simulation methods in the surface energy balance-based remote sensing evapotranspiration models

**Author(s):** *Zhaofei Liu*

It would be greatly appreciated for your kind reviewing to this paper. Thanks very much for your valuable comments and suggestion. For your convenience to re-review the paper, the response corresponding to your comments are described in detail as follows:

*Remote sensing-based surface energy balance typically requires G simulation to close the surface energy balance, which is often a challenge given that G could not be easily sensed from the surface. Hence, most remote sensing-based ET models use an empirical approach to scale G between the two extreme limits of % or fraction of G/Rn within the open surface and full canopy. This % or fraction G/Rn is characterized by vegetation and remotely sensed indices like NDVI, LAI, albedo, LST, etc using simple empirically derived values. This paper aims to study the spatiotemporal variations of this empirical relationship (G, Rn, H) and evaluate some of the remote sensing-based empirical methods using half-hourly global flux observations data. While, I think it is important to improve remote sensing approaches to simulate G, as it will also improve remote sensing and surface energy balance-based ET models, the results and discussion, as presented in the paper, are a little challenging to follow with not many insights into how G simulation in remote sensing-based ET models could be improved. So, I have some major issues (and some minor issues) that the author needs to consider before the paper can be reevaluated.*

*Major Comments*

*The paper is more focused on the assessment of Rn and G relationships than the evaluation of simulated G within the existing remote sensing-based ET models. So I wonder if this should be reflected in the title of the paper, which suggests that the paper is focused on the evaluation of the existing methods. Note that the empirical nature of G simulations and their uncertainty in remote sensing-based ET models is a well-known issue. So while the optimization of regression coefficients (e.g., those in the LC methods) is nice, the finding that the coefficients differ across different parts of the world is obvious. What is more important is to present an idea about how this*

*empiricism and uncertainty can be reduced and a globally applicable model could be developed. The paper falls short on this part.*

**Reply:** Yes. This study is focused on the evaluation of the existing methods. The title has been revised to "Accuracy of five ground heat flux empirical simulation methods in the surface energy balance-based remote sensing evapotranspiration models" to make it more clear. As mentioned in Line 96-99, "This study addresses four key objectives: (1) investigating the temporal and spatial variations and common characteristics of the empirical relationship between G and Rn; (2) evaluating the accuracy of five empirical methods in simulating half-hourly G from Rn; and (3) investigating the performance of five methods at different times during the intra-day and the spatial distribution of simulation accuracy at global flux observation sites." However, it is out of the scope of this study to present an idea about how this empiricism and uncertainty can be reduced and a globally applicable model could be developed. These are issues that model users and developers need to consider more. Results of this paper can provide some references for RS ET data users and the remote sensing evapotranspiration modelers. For example, the applications of RS ET data sets need more caution in tropical regions, and further improvement of G simulations at low-latitude areas and noon periods are recommended for RS ET modelers.

*The paper acknowledges the limitation of existing G derivation methods in remote sensing-based ET models but does not consider some of the widely used approaches, such as the one used in the SEBAL model, which would require albedo and LST. The author acknowledged that SEBAL based G method was found to be working better than other approaches in another study (Saadi et al. 2018). The G models evaluated in this paper are very similar in nature. Hence, it is important to incorporate G models with different structures/inputs. Note that obtaining albedo and LST for these sites is as easy as obtaining NDVI. The author should have incorporated some additional G derivation methods used in the common remote sensing-based ET model.*

**Reply:** The author had used LST data at regional scales, while were not familiar with albedo data. The used LST data is the Terra Moderate Resolution Imaging Spectroradiometer (MODIS) MOD11A1 product, which is produced daily LST at a spatial resolution of 1 km. The evaluation in this study was based on daily data series. For daily series of the global LST dataset, the author only found that the MODIS

MOD11 dataset was available. The MOD11B product provides daily per pixel Land Surface Temperature and Emissivity (LST&E) in a 1,200 by 1,200 kilometer (km) tile with a pixel size of 5,600 meters (m). There are hundreds of files for each day in the dataset (MOD11A and MOD11B) covering the global land. As for 230 flux sites used in this study, each site is needed to be corresponded to these hundreds of files. Huge amounts of data need to be downloaded, i.e. hundreds of files per day multiplied by number of days in the observed daily series of flux sites. In fact, Saadi et al. (2018) only evaluated the methods at a single observed site. The NDVI dataset used in this study is a single file per day with global coverage, and the workload is relatively acceptable. In addition, the authors had tried several methods to download MODIS product but without success at the beginning of this study. It was also failed to download these products in the past few days. This work is beyond the author's capacity. Therefore, the methods embedded with LST data were not evaluated in this study.

*The author mentioned that observed G is taken as the residual of the energy balance to evaluate different G models, assuming that all other components are perfectly derived. While the author acknowledges this in section 4.1 (Line 364-365), I think still problematic because no attempt has been made to address this issue. Here, there is no information on how the energy balance was closed (or was not unclosed) or corrected. The observed G used in this paper and all error metrics presented hence could be highly biased and uncertain.*

**Reply:** It assumes that the measurements of Rn, H and LE are accurate in this study. These measurements might have some errors. However, it is not considered in this study. To the author's knowledge, the eddy covariance measurements of H and LE are generally considered to be the most accurate observations available.

As described in the second paragraph (Line 39-44), ground heat flux (G) is the soil heat flux at the surface. It is difficult to observe directly, due to technical limitations (Wang and Bou-Zeid, 2012; Gao et al., 2017). Soil heat flux (referred to as G') is generally measured using heat flux plates near the surface (within a few millimeters of the surface) in the flux tower observation sites. There were numerous studies investigated on the surface energy balance closure issue at flux sites. The observed G' instead of G was generally used to investigate the energy balance ratio (Wilson et al., 2002). However, the difference between G' and G could be 50% because of the soil

heat storage within the layer from the surface to the flux plate (Heusinkveld, 2004; Yue et al., 2011; Wu et al., 2020). A large error is produced if the soil heat storage is ignored in the G calculation (Meyers and Hollinger, 2004; Lu et al., 2018). The energy balance closure problem might be largely caused by the soil heat storage (Foken, 2008).

Theoretically, surface energy is balanced. The energy unclosure might be mainly caused by the error of the observed data. Compared with G, other energy terms can be observed more accurately. Therefore, the surface energy balance method was used as references in this study. As mentioned in the section of Discussion (Line 362-365), "The eddy covariance measurements of H and LE are generally considered to be the most accurate observations available. The **Eq. (1)** makes full use of the surface energy term that can be accurately measured at present. In other words, it assumes that the measurements of Rn, H and LE are accurate in this study. The uncertainties of measurements are not considered in this study."

The residual of the surface energy balance method has been validated by an experimental site in the West of Spain (van der Tol, 2012).

In other words, it was assumed that Rn is accurately simulated by the RS ET models. Therefore, it should be noted that the uncertainty in Rn calculation was also a source of error in G simulations in ET models."

*It is not clear how the coefficients of the LC methods are calibrated in this study. Are these just the regression coefficients or other optimization methods used? Was any calibration/validation approach used (using independent sets of data)?*

**Reply:** The coefficients of the LC methods are calibrated by the NSE. The author realizes that there are many multi-objective parametric calibration methods. But these methods are too time-cost to be achieved for hundreds of sites. A new sentence "The parameters of these methods were calibrated by the Nash-Sutcliffe efficiency (NSE) at each observation site." is added in Line 148-149.

A new sentence "At each site, daily series of each half-hour were divided into two parts: the first 80% of the data were used for parameter calibration and the rest were used for validation." is added in Line 147-148 to make calibration/validation more clear. In addition, the author tried to test robustness of the methods at some sites. Daily series were randomly assigned to one of two datasets: 80% were assigned to the calibration dataset and 20% to the validation dataset. The process of random assignment was repeated to generate 100 independent datasets. Results showed that these methods are robust. The author would like to add these results in Supplementary Materials if possible.

*I am surprised why the author did not test the actual LC methods (i.e., the original coefficients) used in different ET models considered in this study. In addition, it is important to mention how these different ET models come up with different empirical coefficients.*

**Reply:** This issue had been discussed in the first and second paragraphs of the Section 4.2. As mentioned in Line 394-398, "The LC method is most commonly used in the RS ET models. The coefficients applied to each model were different. The coefficients of the LC method in the TSEB (Norman et al., 1995), ALEXI (Anderson et al., 1997), DisALEXI (Norman et al., 2003), MOD16A2 (Mu et al., 2011), and modified TSEB (Ait Hssaine et al., 2020) ET models were 0.35, 0.31, 0.30, 0.39, and 0.37, respectively. The coefficient of the method in the GLEAM model was 0.05, 0.2 and 0.25 for the tall canopy, short vegetation and bare soil, respectively (Miralles et

al., 2011)." In this study, the parameters of the LC methods were calibrated for each half-hour periods at each site. Results showed that the optimal parameter values varied significantly in different sites and half-hour periods. The author had tested some actual LC methods, and found that the original parameter values could accurately simulate G at some sites, but induced large errors in the G simulations in other regions. Therefore, it is recommended that model developers consider the spatial variations of G simulation parameters in RS ET modeling on a global scale (Line 405-407).

According to your valuable comments, the title has been revised to "Accuracy of five ground heat flux empirical simulation methods in the surface energy balance-based remote sensing evapotranspiration models". In addition, "empirical based" has also been added in the main text. In this study, the parameters of each empirical method were calibrated for each half-hour periods at each site. According to your valuable comments, descriptions of calibration/validation have been added in Line 148-150, as follows "At each site, daily series of each half-hour were divided into two parts: the first 80% of the data were used for parameter calibration and the rest were used for validation. The parameters of these methods were calibrated by the Nash-Sutcliffe efficiency (NSE) at each observation site."

*I find no difference between the contents in the abstract and the conclusion. Both summarize key results with no discussion on the key reasons for differences in model performances and insights into how future remote sensing-based G models can be improved. I couldn't find the main objective of the paper in the abstract.*

**Reply:** The abstract and the conclusion have been revised to avoid repeat problem. In the third paragraph of the section 4.1, it was found that "the accuracy of the G simulation is affected by the correlation between Rn and G." However, the other (physical) reasons for differences in model performances have not been found in this study. It might be caused by the differences in climate, soil and land cover. According to your valuable comments, evaluations of seven land cover types have been added in revised manuscript. Because the observation sites used in this study has a land cover classification. The sites were divided into seven land cover types: Forest, Grassland, Cropland, Wetland, Shrubland, Savanna, and Other types. Figure 3, 4 and 7 (Figure 8 in the revised version) have been revised according to your valuable comments. A

new Figure 7 has been added. Descriptions of these figures have also been added as follows,

[revised manuscript text omitted]

This study is focused on evaluation of five ground heat flux empirical simulation methods in ET models. It only provides some references for ET modelers. For example, consider the spatial variations of G simulation parameters in RS ET modeling on a global scale, and further improvement of G simulations at low-latitude areas and noon periods are recommended.

In Line 10-11, a new sentence "The G simulation methods had been evaluated at many individual sites, while there were relatively few multi-site evaluation studies." has been added to make it clear.

*Minor comments:*

*Line 7: Instead of saying "According to 230 flux site observations" better say Based on the assessment from 230....*

**Reply:** Thanks for your valuable comments. "According to 230 flux site observations" has been revised to "Based on the assessment from 230 flux site observations".

*Line 8-9: Based on the previous statement, it shows that G accounts for a significant proportion of the daily surface energy balance.*

**Reply:** Yes. It used "important role" to describe this issue.

*Line 19: It's not the accuracy of the sites. It's rather the accuracy of the models in these sites.*

**Reply:** According to your valuable comments, this sentence has been revised to "The accuracy of the model was generally higher in Northern Hemisphere sites than in Southern Hemisphere sites."

*Line 31-42: It's better to differentiate "ground heat flux" or "soil heat flux" by providing their physical meanings and with more detailed descriptions. The author defines soil heat flux as the heat flux measured by the flux plates near the surface.*

**Reply:** Yes.

*Line 74-75: Suggest citing Roerink et al., 2000 and Merlin et al., 2014 right after the corresponding model names*

**Reply:** Thanks very much for your valuable comments. This sentence has been revised to "The solutions of G in the first two models were also applied to the Simplified Surface Energy Balance Index (S-SEBI) (Roerink et al., 2000) and Four-source Surface Energy Balance (SEB-4S) (Merlin et al., 2014) models, respectively."

*Lines 101-110: Given the numbers of towers from different networks, could you please indicate how you came up with the number "230" (i.e., 230 sites used in this study).*

**Reply:** There were 189 FLUXNET2015 sites and 60 FLUXNET-CH4 sites were used in the analysis. There were 19 sites belonging to both FLUXNET2015 and FLUXNET-CH4. Four sites obtained from the TERN OzFlux dataset were also included in FLUXNET products. Therefore, 230 sites used in this study.

The sentence "There were 19 sites belonging to both FLUXNET2015 and FLUXNET-CH4, and flux observation data from four sites in Australia were obtained from the TERN OzFlux dataset, which was a long and continuous series up to 2019 (Beringer et al., 2016)." has been revised to "There were 19 sites belonging to both FLUXNET2015 and FLUXNET-CH4. Flux observation data from four sites in Australia were obtained from the TERN OzFlux dataset. These four sites were included in FLUXNET products, but were with a longer and continuous series up to 2019 (Beringer et al., 2016)." to avoid misunderstanding.

*Line 270-272: I do not think you can say NSE is suitable but RE and KGE for evaluation. Yet, you are using RE, RMSE, and KGE for model evaluation. Maybe you need to rephrase the sentence. It is better to justify the choice of model evaluation metrics in the methods section.*

**Reply:** The sentence "The evaluation of the model in this study included four criteria." has been revised to "In this study, four criteria were tried to evaluate the model." In addition, in Line 150, the sentence "The criteria used to evaluate these simulations included…" has been revised to "The criteria tried to evaluate these simulations included".

*Line 340: Please mention the optimization process in the Methods section*

**Reply:** Descriptions of the parameter calibration have been added in Line 152-154, as follows, "At each site, daily series of each half-hour were divided into two parts: the first 80% of the data were used for parameter calibration and the rest were used for validation. The parameters of these methods were calibrated by the Nash-Sutcliffe efficiency (NSE) at each observation site."

*Line 329: How can daily G be simulated at 6:30? Shouldn't this be G only or half-hourly G?*

**Reply:** The sentence "The LC method accurately simulated daily G of most sites at 6:30" has been revised to "The LC method accurately simulated G at 6:30 in most sites" to make it clear. In addition, similar revisions have also been made in Line 232, 234, 243, and 339.

*Line 383: MODIS is not used at 10:30 and 13:30. MODIS data represents conditions around these times.*

**Reply:** Yes. Thanks for your valuable comments, this sentence has been revised to "For example, MODIS data represents conditions around 10:30 and 13:30".

*Line 393-398: redundant information in the paper*

**Reply:** Yes. These sentences "The LC method is most commonly used in the RS ET models. The coefficients applied to each model were different. The coefficients of the LC method in the TSEB (Norman et al., 1995), ALEXI (Anderson et al., 1997), DisALEXI (Norman et al., 2003), MOD16A2 (Mu et al., 2011), and modified TSEB (Ait Hssaine et al., 2020) ET models were 0.35, 0.31, 0.30, 0.39, and 0.37, respectively. The coefficient of the method in the GLEAM model was 0.05, 0.2 and 0.25 for the tall canopy, short vegetation and bare soil, respectively (Miralles et al., 2011)." have been deleted.

*Line 416-417: These data are easy to get. It may not be a good idea to ignore Bastiaanssen (1995) Method when it was found to be working better than other approaches in another study (Saadi et al. 2018).*

**Reply:** This has been explained in the reply of the second Major Comment. The author has tried hard to download LST data, but failed.

*Line 420: The difference among different methods was not significant because NDVI and fc are highly correlated (in fact NDVI is likely used to derive fc) and they are calibrated similarly.*

**Reply:** Yes. The author agrees with that. But the performance of the different methods varied at some sites.

*Line 430: there may be a case when a large error in G may be canceled by a large error in Rn leading to reasonable estimates of available energy (Rn-G), which is*

*further partitioned into sensible and latent heat fluxes.*

**Reply:** Yes. The author agrees with that. This sentence has been revised to "A large error in the G simulation might be induced in the ET modelling process, thereby reducing the accuracy of the ET estimates."

---

## Author Comment (AC2)

**Response (Referee #2 comment)**

**Ms. Ref. No.**: hess-2022-125

**Revised title**: Accuracy of five ground heat flux empirical simulation methods in the surface energy balance-based remote sensing evapotranspiration models

**Author(s):** *Zhaofei Liu*

It would be greatly appreciated for your kind reviewing to this paper. Thanks very much for your valuable comments and suggestion. For your convenience to re-review the paper, the response corresponding to your comments are described in detail as follows:

*This paper analyzes the relationship between G and Rn at a continental scale with hundreds of flux site measurements. This work is interesting to RS energy balance ET model users. It concluded that the linear coefficient (LC) method and the methods embedded with the normalized difference vegetation index (NDVI) were able to accurately simulate a half-hourly G series at most sites. The methods using fractional vegetation coverage showed poor performance. The highest accuracy was exhibited during sunrise periods (6:00-7:00), followed by sunset periods (17:00-18:00). The lowest accuracy was observed at noon periods (10:00-15:30). These conclusions are important for RS ET simulation. From this point, this work deserves a publication on HESS. Meanwhile, it also has some shortages which needs more clarification. The following are some comments.*

*Two major comments:*

*G was taken as the residual of Rn-H-LE in this paper, without considering the energy balance issue. This method might work for some low canopies which has a relative homogeneous land surface. The measurement of H and LE might have problem for forest site, since H and LE sensor are not high enough to be out of the sub-roughness layer on the canopy top. Hereby, this paper needs some discussion on why the energy unbalance item can be all partitioned to G, or what kind of data quality controlling process can make him/her believe that H and LE measurement at the selected sites are accurate and they don`t need energy balance correction.*

**Reply:** Thanks for your valuable comments. The observation sites used in this study has a land cover classification. The sites were divided into seven land cover types: Forest, Grassland, Cropland, Wetland, Shrubland, Savanna, and Other types. Evaluations of seven land cover types have been added in revised manuscript. The

low performance in some Forest sites might be due to the fact that the H and LE sensor are not high enough to be out of the sub-roughness layer on the canopy top as you mentioned.

As described in the second paragraph of the Introduction section, "G, which is the soil heat flux at the surface, is difficult to observe directly, due to technical limitations (Wang and Bou-Zeid, 2012; Gao et al., 2017), and direct estimation of G using RS data is not possible (Kalma et al., 2008; Allen et al., 2011; Saadi et al., 2018)." Therefore, as discussed in the first paragraph of the Discussion section, "The eddy covariance measurements of H and LE are generally considered to be the most accurate observations available. The Eq. (1) makes full use of the surface energy term that can be accurately measured at present. In other words, it assumes that the measurements of Rn, H and LE are accurate in this study."

*Eq.2-6, the author has optimized a, a1, a2, and b. However, they did not analyze the values of these optimized variable. Figure 8 only show optimized values for three methods, without show other two methods. a1 and a2 in eq. 5 has their definition or physical meaning in the original publication. Whether the optimized values for these two parameters still follow the range of their physical meaning? I suggest to do some statistical analysis of these optimized parameter values. This can help other users when using equation 2-5. Chen et al. 2019 AFM has optimized fc based G/Rn equation. Please make a comparison with this study. They have optimized a1, a2 with a classification of land covers and canopy types. Since these parameter values could varies due to canopy covers, I suggest this paper also use canopy classification to analyze the NSE values in figure 6, KGE, RMSE, RE in figure 5, R^2 and slope in figure 3. Figure 1 can be also divided into different land covers. And, please also conclude which of the five methods is the best for which land covers or canopy classification. This result will be more useful for the RS ET model users. Figure 4, it would be interesting to analyze the linear fitting R^2 between G/Rn and NDVI for different canopy. The same problem with figure 7. Figure5, please also add Re, RMSE and KGE for other methods, not only show the LC method.*

**Reply:** According to your valuable comments, the evaluations of seven land cover types have been added in revised manuscript. Figure 3, 4 and 7 (Figure 8 in the revised version) have been revised according to your valuable comments. A new

Figure 7 has been added. Descriptions of these figures have also been added as follows,

[revised manuscript text omitted]

Line 480-481, a new sentence "This has also verified by Chen et al. (2019)." was added to make it clear.

In Figure 5, it was focus on some problems about the KGE, RMSE and RE in evaluating the model performance at different sites and time periods. However, the author would like to provide the land cover results of the KGE, RMSE and RE in the

Supplementary Materials if possible.

*Some minor comments:*

*Figure 6. The NSE value is calculated after or before a, a1, a2, b were optimized? The figure description should include this information.*

**Reply:** Yes. The NSE value is calculated after the parameters were optimized. The figure title has been revised to "Figure 6: The NSE simulated by the (a) LC, (b) LC_NDVI_P, (c) LC_NDVI_E, (d) LC_fc_SE and (e) LC_fc_ST methods based on optimized parameters in each site and half-hour intervals." to make it clear.

*Figure 8, the label for y-axis is not accurate, please revise it.*

**Reply:** Yes. The label for y-axis in Figure 8 (Figure 9 in revised version) has been revised.

*Figure 1a shows that G and Rn has a time phase difference in their diurnal variation. However, this paper does not consider this effect. Please explain why not consider this effect in their using G/Rn equations.*

**Reply:** Yes. There is a time phase difference in the diurnal variation of G and Rn. The time phase difference varied at different sites. This effect has been reduced by parameter optimization at each site and half-hour period.

*These ET datasets include, but are not limited to, the Breathing Earth System Simulator (BESS) (Jiang and Ryu, 2016), Moderate Resolution Imaging Spectroradiometer (MODIS; MOD16A2) (Mu et al., 2011), GLEAM (Miralles et al., 2011), and Numerical Terradynamic Simulation Group (NTSG) (Zhang et al., 2010) products. There are more global ET products which is based on energy balance method, such as EB-ET (Chen et al. 2021), http://data.tpdc.ac.cn/zh-hans/data/df4005fb-9449-4760-8e8a-09727df9fe36/?q=energy%20balance. This ET product is based on energy balance method. The author may think that this study is more useful for energy balance based ET models.*

**Reply:** This sentence has been revised to "These ET datasets include, but are not limited to, the Breathing Earth System Simulator (BESS) (Jiang and Ryu, 2016), Moderate Resolution Imaging Spectroradiometer (MODIS; MOD16A2) (Mu et al., 2011), GLEAM (Miralles et al., 2011), Numerical Terradynamic Simulation Group

(NTSG) (Zhang et al., 2010) and Thermal Energy Balance (Chen et al., 2021) products."

*The surface energy balance method provides an alternative solution for assessing the G simulation schemes (van der Tol et al., 2012). This method could avoid the inconsistent spatial scale of G with that of LE and H in field measurements. I don`t understand what`s the meaning of these two sentences, please rephrase them.*

**Reply:** As mentioned in Line 85-93, the gradient and calorimetry approaches had been used for evaluations of G simulations. These evaluations were limited to a single site scale because field observations of soil thermal properties were available only at a few sites. Therefore, the surface energy balance method provides an alternative solution for assessing the G simulation schemes (van der Tol et al., 2012). And this method could avoid the inconsistent spatial scale of G with that of LE and H in field measurements.

*The slope and R2 of the linear fitting curve were -0.012 and 0.92, respectively. Are you sure the slope is negative value?*

**Reply:** Yes. As shown in Figure 2-c, the slope of the linear fitting curve for mean G/Rn of all sites in the daytime periods is -0.012.

*Change "use Rn to calculate G in the RS inversion of ET" to use Rn to calculate G in RS based energy balance ET models (Chen et al. 2019 AFM; Chen et al. 2021 JGR).*

**Reply:** "use Rn to calculate G in the RS inversion of ET" has been revised to "use Rn to calculate G in the RS based energy balance ET models".

*Some references about energy balance ET models should be cited:*
*Chen, X., et al. (2019). "Optimization of a remote sensing energy balance method over different canopy applied at global scale." Agricultural and Forest Meteorology 279: 107633.*
*Chen, X., et al. (2021). "Remote Sensing of Global Daily Evapotranspiration based on a Surface Energy Balance Method and Reanalysis Data." Journal of Geophysical Research: Atmospheres 126(16): e2020JD032873.*
*Chen, X., et al. (2014). "Development of a 10-year (2001–2010) 0.1° data set of land-surface energy balance for mainland China." Atmos. Chem. Phys. 14(23):*

*13097-13117*

**Reply:** These references have been cited in revised manuscript.

---

## Author Comment (AC3)

**Response (Referee #3 comment)**

**Ms. Ref. No.**: hess-2022-125

**Revised title**: Accuracy of five ground heat flux empirical simulation methods in the surface energy balance-based remote sensing evapotranspiration models

**Author(s):** *Zhaofei Liu*

It would be greatly appreciated for your kind reviewing to this paper. Thanks very much for your valuable comments and suggestion. For your convenience to re-review the paper, the response corresponding to your comments are described in detail as follows:

*The main target of this paper is to test several empirical formulations of the ratio between the soil heat flux G and the net radiation Rn, which is a key issue for estimating evapotranspiration through surface energy budget models forced by instantaneous remote sensing surface temperature data.*

*Main issues with the paper are:*

*The evaluation dataset is based on the sole estimate of G as a residual of the energy budget from flux tower measurements; G being usually small compared to the turbulent fluxes, the total uncertainty is high, and a more robust method would have been to do, as classically done, a correction of the subsurface sol heat flux plates measurements, with potentially a further correction with the residual G estimate, bearing in mind that turbulent fluxes are generally underestimated. Furthermore, the FLUXNET dataset is not representative of the agro- eco-types where remotely sensed ET estimates are required; especially, crops in Mediterranean and semi-arid climates are largely underrepresented. This limits the study's impact.*

**Reply:** As described in Line 36-38 (40-42 in revised manuscript), "Over bare soils or sparsely vegetated surfaces, G can reach half of the net radiation (Rn) (Heusinkveld et al., 2004). Even under full vegetation cover, G is significant, especially when turbulent processes are less active (Gentine et al., 2012)."

As described in Line 48-60 (52-64 in revised manuscript), "There are numerous schemes for estimating G (Wang and Bou-Zeid, 2012; Gao et al., 2017; Wu et al., 2020)…". "However, applications of these physical mechanism-based approaches are restricted to only a few sites, due to the limitations of field observations of soil thermal properties (Mayocchi and Bristowa, 1995; Kustas et al., 2000). Soil thermal properties are affected by soil texture, mineralogical composition, bulk density, and the surrounding environment (e.g., soil moisture and temperature) (Peng et al., 2017; Ju and Hu, 2018). In other words, soil thermal properties vary with time and space."

"To estimate ET in RS models, G is usually obtained from empirical relations with Rn." In this study, accuracy of five ground heat flux empirical simulation methods in the surface energy balance-based remote sensing evapotranspiration models was evaluated by flux site observations.

The observation sites used in this study has a land cover classification. The sites were divided into seven land cover types: Forest, Grassland, Cropland, Wetland, Shrubland, Savanna, and Other types. It represents different agro-eco-types. According to your valuable comments, evaluations of seven land cover types have been added in revised manuscript. Figure 3, 4 and 7 (Figure 8 in the revised version) have been revised according to your valuable comments. A new Figure 7 has been added. Descriptions of these figures have also been added as follows,

[revised manuscript text omitted]

*The number of empirical equations under study is limited, esp. regarding previous works (Sun et al., 2013\*, Bonsoms and Boulet 2022\*\*)*

**Reply:** These works have been cited in revised manuscript. The author has reviewed these references carefully, and found that the empirical equations missed in this study are some methods required albedo and LST data. As described in Line 473-476 in revised manuscript, these equations were not evaluated in this study due to data limitations.

The author had used LST data at regional scales, while were not familiar with albedo data. The used LST data is the Terra Moderate Resolution Imaging Spectroradiometer (MODIS) MOD11A1 product, which is produced daily LST at a spatial resolution of 1 km. The evaluation in this study was based on daily data series. For daily series of the global LST dataset, the author only found that the MODIS MOD11 dataset was available. The MOD11B product provides daily per pixel Land Surface Temperature and Emissivity (LST&E) in a 1,200 by 1,200 kilometer (km) tile with a pixel size of 5,600 meters (m). There are hundreds of files for each day in the dataset (MOD11A and MOD11B) covering the global land. As for 230 flux sites used in this study, each site is needed to be corresponded to these hundreds of files. Huge amounts of data need to be downloaded, i.e. hundreds of files per day multiplied by number of days in the observed daily series of flux sites. In fact, Saadi et al. (2018) only evaluated the methods at a single observed site. The NDVI dataset used in this study is a single file per day with global coverage, and the workload is relatively acceptable. In addition, the authors had tried several methods to download MODIS product but without success at the beginning of this study. It was also failed to download these products in the past few days. This work is beyond the author's capacity. Therefore, the methods embedded with LST data were not evaluated in this study.

*I am concerned with Figure 1a: H and Rn are equal ! Also, why are the flux values so low for half hourly flux estimates ? Some explanation is required here; if G is the residual, the energy budget is closed, the SEB average of all sites should also be closed for each half hourly value, i.e Rn-G=H+LE. Also, G' seems to be an uncorrected G measurement at a few cm depth (please confirm, G' is actually not*

*defined properly in the paper), the corrected G' at the surface should be shown and analysed for all sites compared to G, esp. since the normalized (G) and (G') looks similar (1e versus 1f).*

**Reply:** Figure 1a includes the primary and secondary y-axes. In Figure 1a, the primary and secondary y-axes represent the H and Rn, respectively. The H is very different from Rn. For example, the greatest H value ($<150$ W/m$^2$) accounts for only 40% of the highest Rn value (Figure 1a). The half hourly flux values shown in Figure 1 are calculated from the FLUXNET observations. G is the residual in this study.

As described in the second paragraph of the Introduction section, G' is soil heat flux measurement at a few cm depth. G is the soil heat flux at the surface, which is difficult to observe directly due to technical limitations (Wang and Bou-Zeid, 2012; Gao et al., 2017), and direct estimation of G using RS data is not possible (Kalma et al., 2008; Allen et al., 2011; Saadi et al., 2018). There are too many sites (230) used in this study, it is impossible to show intra-day distribution of flux values for each site. Therefore, the mean flux values of all sites were shown in Figure 1. Yes, the intra-day distribution characteristics of normalized G and G' are similar (1e and 1f). It could also be found that the normalized H and LE are also similar (1c and 1d). All of these intra-day distributions of fluxes are determined by the Rn. In fact, the intra-day distribution of these fluxes is also similar to the Rn (1b).

*Detailed comments:*
*Line 7: what is the difference between "intra-day" and "diurnal" ?*

**Reply:** "diurnal" was expected to describe of or belonging to or active during the daytime. It has been revised to "daytime" to avoid misunderstanding.

*Line 9: add that G is required for RD ET models based on the SEB forced by radiative surface temperature (it is of no importance for other models).*

**Reply:** This sentence has been revised to "This indicates that G plays an important role in remote sensing (RS) energy balance based evapotranspiration (ET) models." according to your valuable comments.

*Line 9: add "empirical", i.e. "G empirical estimation methods"*

**Reply:** "empirical" has been add in revised manuscript, including the Title, Line 10, 11, 31, 86, 103, and 462.

*Line 13: "the two methods ... ": revise the sentence ; I find a bit contradictory that calibrated G/Rn based on NDVI and fractional cover have contrasted performances.*

**Reply:** This sentence has been revised to "The linear coefficient (LC) method and the two methods embedded with the normalized difference vegetation index (NDVI) were able to accurately simulate a half-hourly G series at most sites."

*L65 to 77: all models based on forcing SEB with land surface temperature need an estimate of G/Rn, no need to review them all, better provide an updated review of all G/Rn equations*

**Reply:** Four new references have been added in this paragraph. A new sentence has been added at the end of this paragraph, "More G empirical estimation methods could be found in Sun et al. (2013) and Bonsoms and Boulet (2020)."

*Line 140: we can't use only calibrated parameters for operational applications (i.e. satellite products) so it is important to also test the default (published) parameter values (comment also made by other reviewers).*

**Reply:** The default parameter values had been evaluated in this study. As described in the second paragraph of the section 4.2, "The fixed parameters might be suitable for some regions, but not on a global scale. This study confirmed that the optimal parameter values vary significantly from site to site." In addition, as revised in the last paragraph of this section, "…the optimal values of the model parameters differed among the different sites. This has also verified by Chen et al. (2019)." The author found that the default parameter values published in the references were also optimized by some observations sites data. Therefore, it is recommended that model developers consider the spatial variations of G simulation parameters in RS ET modeling on a global scale.

*Line 370: NO, Santanello and Friedl (2003) do NOT need LST*

**Reply:** Thanks for your valuable comments. It includes a variable "t" in the equation (4) of this reference. The author made a mistake on that. The sentence "However, it requires intra-day land surface temperature (LST) data series, which cannot be obtained by RS. Because RS can only monitor instantaneous LST when a satellite overpasses, it cannot obtain intra-day LST data series." has been deleted.

*Line 420: I don't understand this sentence*

**Reply:** Line 420, "The results of this study indicate that the performance of the different methods varied at some site scales." As described in Line 417-420, Saadi et al. (2018) found the accuracy of three methods was different at an observation site. The sentence in Line 420 means that the performance of methods evaluated in this study is also different at some sites. This sentence has been revised to "The results of this study indicate that the performance of the different methods varied at some sites." to make it clear.

*\* Sun, Z., Gebremichael, M., and Wang, Q.: Evaluation of Empirical Remote Sensing-Based Equations for Estimating Soil Heat Flux, Journal of the Meteorological Society of Japan, 91, 627-638, 10.2151/jmsj.2013-505, 2013.*
*\*\* Bonsoms, J., and Boulet, G.: Ensemble Machine Learning Outperforms Empirical Equations for the Ground Heat Flux Estimation with Remote Sensing Data, Remote Sensing, 14, 1788, 10.3390/rs14081788, 2022.*

---

## Referee Report (RR1)

This manuscript evaluates several empirical methods to estimate ground heat flux, which is interesting, but lack of creativity. Similar studies and conclusions could be found, e.g., Purdy et al. (2016) evaluated the soil heat flux at 88 sites globally based on FLUXNET2015 dataset. The large uncertainty in G estimation at global scale using empirical methods (including those evaluated in this manuscript) was clearly concluded by many previous studies. It's commonly agreed that these empirical equations should be carefully calibrated when applying them to new regions or sites. The site-to-site parameter optimization of empirical method in this study is helpful, but no significant added-values and explicit ideas/suggestions to the community on how to improve the algorithm and accuracy in G estimation at global scale. The author did quite a lot but yet superficial data analysis to show the relationship between G and Rn (and H) without insight interpretation or discussions on the mechanism behind the results, which did not bring sound scientific significance and inspirations to readers.

**Major issues:**

Even though eddy covariance (EC) measurements are widely used, the uncertainty of EC measurements of turbulent fluxes should be carefully evaluated. Actually, the random uncertainty of the measured latent heat flux (LE) by EC could reach 16%, and the random uncertainty of sensible heat flux (H) could reach 18%. Considering the magnitude of G is much smaller than LE and H, the uncertainty of G estimated by SEBR method would be very large, even larger than the magnitude of G itself.

The footprint of net radiation and soil heat flux observations are significantly different from that of H/LE observations. Large uncertainty is anticipated when estimating G directly using the surface energy balance residual (SEBR) method (Eq.1 in the manuscript). Energy imbalance has been an issue in the ground measurements for long time. The author does not seem to have assessed and corrected the energy imbalance at the flux tower sites before using the data, which could be the reason that the author has obtained very large G based on the SEBR method at site level and thus would bring unreliable relationships between G and Rn.

It seems some problems when the author processes the FLUXNET2015 data. According to my experience in the data-screening with FLUXNET2015 data, there are many sites cannot provide G observation (e.g., PA-SPn, and some others), and these sites should be eliminated from the analysis. But these sites were also listed in the Supplementary Table1. The author needs to check it more carefully.

The diurnal variations in the averaged fluxes of the surface energy balance as shown in Fig. 1 seems too smooth to me. I cannot believe these curves come from actual measurements, they are just too perfect.

Why the linear models with NDVI perform better than the model with Fc? The author should give explanation more clearly. Eq. 2 and Eq. 3 are almost the same, it does not

make sense to take them as two different models.

Indeed, the different performance of G estimation in different time (hours of a day) are closely related to the time-lag between G and Rn, which is important but not well explained.

At large scale application using satellite data (NDVI), the author has used the NDVI from AVHRR product with the spatial resolution of 0.05×0.05° which is too coarse to compare with the footprint of ground measurements. This is particularly important for sites where the land surface is heterogeneous around the 0.05×0.05° spatial domain.

In the Discussion section the author state that "it requires intra-day land surface temperature (LST) data series, which cannot be obtained by RS. Because RS can only monitor instantaneous LST when a satellite overpasses, it cannot obtain intra-day LST data series", this is not true! Geostationary satellites can provide LST observations at 15min – 30min intervals.

**Minor issues:**

In Figure 1, explanations for sub-figures are missing in the caption.

Figure 8: I do not understand how to get the median NSE value of each site. Shouldn't it be a single NSE value per site?

It's inappropriate to define 6:00-7:00 as sunrise periods and 17:00-18:00 as sunset periods globally, since the sunrise and sunset time vary with locations and seasons.

Line 190: **"**During data processing, data points with absolute values greater than 10 in the G/Rn or G/H daily series of each period were deleted."
Threshold 10 seems too large, can G be larger than Rn ?
Similar problem is in the range of coefficient α in Eq.1, the maximum value of 1.5 will lead to G is 1.5 times larger than Rn, does it have physical meaning?

In Figure 4, analysis is also done for monthly temporal scale without explaining why it is needed.

---

## Author Response (AR2)

**Author's response**

**Ms. Ref. No.**: hess-2022-125

**Title**: Accuracy of five ground heat flux empirical simulation methods in the surface energy balance-based remote sensing evapotranspiration models

**Author(s):** *Zhaofei Liu*

It would be greatly appreciated for your kind reviewing to this paper. Thanks very much for your valuable comments and suggestion. For your convenience to re-review the paper, the response corresponding to your comments are described in detail as follows:

═══════════════════════════════════════════════════════════

**Responses to Editor' Comments**

*Comments to the author:*

*Dear author,*

*Thank you very much for your responses to review comments.*

*I have obtained further comments on the current revision and would like to invite you to study the comments carefully and submit a new revision.*

*The main issues to be addressed are:*

*1) What are the significantly added-values of your study compared to previous similar studies? You are invited to explicitly propose ideas and suggestions to the community on how to improve the algorithm and accuracy in G estimation at global scale.*

*2) Based on your interpretation and discussions can you identify the mechanisms behind the results, that may be investigated in future research to advance research in land surface energy balance.*

*I hope these comments are useful for improving the quality of your manuscript.*

*Best wishes*

*Bob Su*

*Editor HESS*

\*\*\*\*\*\*\*\*\*\*\*\*\*\*\*\*\*\*\*\*\*\*\*\*\*\*\*\*\*\*\*\*\*\*\*\*\*\*\*\*\*\*\*\*\*\*\*\*\*\*\*\*\*\*\*\*\*\*\*\*\*\*\*\*\*\*\*\*\*\*\*\*\*\*\*\*\*

**Reply 1):** The added-values of this study compared to previous similar studies were described in Line 90-106. It includes two points: (1), previous studies were largely limited to a single site scale, while this study focused on a global multi-site scales. (2), Purdy et al. (2016) evaluated six empirical methods at global scale. However, this

study evaluated G simulation against G' observations. As described in Line 45-54, there is a large difference between G' and G. We used the surface energy balance method to assess the G simulation methods. This method could avoid the inconsistent spatial scale of G with that of LE and H in field measurements, and makes full use of the surface energy term that can be accurately measured at present.

[revised manuscript text omitted]

In addition, the author also found that seasonal variations of G and Rn were also an important factor affecting the accuracy of empirical methods in simulating G. The new sentences have been added in Line 457-461, "The sites with satisfied model performance were generally characterized by seasonal variation in G and Rn due to

the climate in these regions. Conversely, the sites with poor model performance showed little seasonality. Whether G and Rn have seasonal variations was also an important factor affecting the accuracy of empirical methods in simulating G. This was consistent with the evaluation results of the LE simulation accuracy (Liu, 2021; 2022)."

Please find details in the revised manuscript with tracks.

**Responses to Reviewer #4 Comments**

*This manuscript evaluates several empirical methods to estimate ground heat flux, which is interesting, but lack of creativity. Similar studies and conclusions could be found, e.g., Purdy et al. (2016) evaluated the soil heat flux at 88 sites globally based on FLUXNET2015 dataset. The large uncertainty in G estimation at global scale using empirical methods (including those evaluated in this manuscript) was clearly concluded by many previous studies. It's commonly agreed that these empirical equations should be carefully calibrated when applying them to new regions or sites. The site-to-site parameter optimization of empirical method in this study is helpful, but no significant added-values and explicit ideas/suggestions to the community on how to improve the algorithm and accuracy in G estimation at global scale. The author did quite a lot but yet superficial data analysis to show the relationship between G and Rn (and H) without insight interpretation or discussions on the mechanism behind the results, which did not bring sound scientific significance and inspirations to readers.*

**Reply:** The added-values of this study compared to previous similar studies were described in Line 90-106. It includes two points: (1), previous studies were largely limited to a single site scale, while this study focused on a global multi-site scales. (2), Purdy et al. (2016) evaluated six empirical methods at global scale. However, this study evaluated G simulation against G' observations. As described in Line 45-54, there is a large difference between G' and G. We used the surface energy balance method to assess the G simulation methods. This method could avoid the inconsistent spatial scale of G with that of LE and H in field measurements, and makes full use of the surface energy term that can be accurately measured at present.

In Line 98-100, new sentences "Purdy et al. (2016) evaluated six empirical methods of G simulation against G' observations at 88 flux sites. This was a very meaningful study on a global scale. However, there is a large difference between G' and G which has been described above." have been added to make it more clear.

The added-values of this study and explicit suggestions for improving the accuracy in G estimation include several points:

(1) The evaluated flux sites are much more than previous studies.

(2) It investigates temporal and spatial accuracy of empirical methods of G simulation. The evaluation results are expected to provide reference for RS ET model application

and developers. We find that the performance of these methods was good and poor at some sites and time periods and some land-cover types. RS ET modelers could check the advantage of the models at good performance regions, and find why the models are poor at some other areas, then revise the models to improve the accuracy at poor performance regions.

(3) Empirical methods of estimating G from Rn were evaluated in this study. Results show that the simulation accuracy is mainly affected by the correlation between G and Rn. As added new sentences in Line 540-544, "the weak correlation between the G and Rn is the physical reason for the poor accuracy of G simulation in these regions and sites. Instead of the Rn, finding another variable that has a physical connection and strong correlation with the G might be a more efficient solution to improve the accuracy of the empirical estimation method for G."

(4) The optimal parameter value of each method varied significantly at different sites. Therefore, the fixed parameter values in the G simulation methods do not match the actual situation. Variable parameters are recommended in empirical methods of G simulation to improve accuracy.

The physical mechanism behind the evaluation results of the simulation accuracy of empirical methods is mainly affected by the correlation between G and Rn. Therefore, before the evaluation of empirical methods (Section 3.3), the Section 3.2 is "Temporal and spatial analysis of the empirical relationship between G and Rn and between G/Rn and NDVI".

The Abstract, Introduction, Discussion and Conclusion sections have been revised according to your valuable comments. Please find details in the revised manuscript with tracks.

*Major issues:*

*Even though eddy covariance (EC) measurements are widely used, the uncertainty of EC measurements of turbulent fluxes should be carefully evaluated. Actually, the random uncertainty of the measured latent heat flux (LE) by EC could reach 16%, and the random uncertainty of sensible heat flux (H) could reach 18%. Considering the magnitude of G is much smaller than LE and H, the uncertainty of G estimated by SEBR method would be very large, even larger than the magnitude of G itself.*

**Reply:** At present, there are uncertainties in turbulent fluxes observation, including EC measurements of flux variables. However, as described in Line 426-429, "The

eddy covariance measurements of H and LE are generally considered to be the most accurate observations available, and have been widely used as a reference to evaluate other simulation methods. The Eq. (1) makes full use of the surface energy term that can be accurately measured at present. In other words, it assumes that the measurements of Rn, H and LE are accurate in this study. The uncertainties of measurements are not considered in this study." The estimation of EC measurements uncertainty is out of the scope of this study.

*The footprint of net radiation and soil heat flux observations are significantly different from that of H/LE observations. Large uncertainty is anticipated when estimating G directly using the surface energy balance residual (SEBR) method (Eq.1 in the manuscript). Energy imbalance has been an issue in the ground measurements for long time. The author does not seem to have assessed and corrected the energy imbalance at the flux tower sites before using the data, which could be the reason that the author has obtained very large G based on the SEBR method at site level and thus would bring unreliable relationships between G and Rn.*

**Reply:** Yes. As described in Line 53-54, "The spatial scale of the G observation is also much smaller than that of the H and latent heat flux (LE) estimates (Shao et al., 2008; Verhoef et al., 2012)." Because the footprint of net radiation and soil heat flux observations are significantly different, the net radiation, H and LE observations with relatively similar footprint are used to calculate surface soil heat flux with corresponding spatial scale in this study. It makes full use of the advantages of relatively consistent footprint scale of the surface energy term (Rn, LE, and H).

Energy balance is a universal principle. The energy terms on the surface are also in balance. The surface energy imbalance, which is calculated from the observed data of the local surface energy term, might be mainly caused by the error of observation data. The uncertainty in measuring surface soil heat flux (G) includes: 1) the significant difference between the footprint of soil heat flux observation and other surface energy terms (e.g. Rn, LE, and H); 2) and a large difference between G and soil heat flux measured using heat flux plates near the surface, which is described in the second paragraph of the Introduction section (Line 45-54). These uncertainties of G measurements are important reasons for the imbalance of observed surface energy.

The mean value of G is definitely lower than the average of Rn. As described in the first paragraph of the section 3.1 (Line 175-178), based on the mean of 230

FLUXNET sites, LE, H, and G accounted for 34.5%, 46.3%, and 19.2% of Rn, respectively. G accounted for 28.8% of Rn when only daytime periods were considered. However, the daily or hourly value of G might be several times that of Rn. This is not rare in flux observations from 230 FLUXNET sites. For example, it is possible that G is larger than Rn on cloudy days or at night.

*It seems some problems when the author processes the FLUXNET2015 data. According to my experience in the data-screening with FLUXNET2015 data, there are many sites cannot provide G observation (e.g., PA-SPn, and some others), and these sites should be eliminated from the analysis. But these sites were also listed in the Supplementary Table1. The author needs to check it more carefully.*

**Reply:** Thanks very much for your valuable comments. There are 212 FLUXNET2015 sites and 81 FLUXNET-CH4 sites available from https://fluxnet.org/. However, there are many sites lack of measuring LE, H, Rn, and G'. The first three variables were used for simulating G in this study. Therefore, 230 sites with measurements of LE, H, Rn were used. These sites were listed in the Supplementary Table1. There were 167 sites observing G'. G' data series were only used in the Section 3.1 to identify intra-day distribution characteristics of observed surface energy balance items.

As described in Line 118-120, "All missing values were eliminated. For example, if there were missing values on a certain day, all data on that day were discarded. Therefore, only days with fully available half-hourly data were used in the analysis. Only sites with a data series longer than 360 days were used." In addition, there are also several problems in the raw dataset. For example, there are continuous identical values in Rn measurements at some sites. These continuous identical values were treated as missing values in this study. There are also many missing values in NDVI data covered flux sites. Overall, these data had been carefully preprocessed by the author. It cost a lot of time and energy.

According to your valuable comments, revisions are as follows,

Line 122, the new sentence "G' was not observed at 63 sites (**Table S1**)." was added.

Line 175-177, the sentence "**Figure 1** shows the intra-day distribution of half-hourly Rn, H, LE, G, and G', derived from the mean of 230 FLUXNET sites." has been revised to "**Figure 1** shows the intra-day distribution of half-hourly Rn, H, LE, G, and

G'. The first four variables and G' were derived from the mean of 230 and 167 FLUXNET sites (**Table S1**), respectively."

In Supplementary Materials, Table S1, the sites lack of G' measurements have been marked by "*". A new sentence "Note: * represents the soil heat flux (G') is not observed." has been added.

*The diurnal variations in the averaged fluxes of the surface energy balance as shown in Fig. 1 seems too smooth to me. I cannot believe these curves come from actual measurements, they are just too perfect.*

**Reply:** The Fig. 1 is the diurnal variations of surface energy balance in the averaged values from multi-sites. The Rn, H, LE, and G were derived from the mean of 230 sites, while G' is calculated from the mean of 167 sites. The author could provide the raw data for this figure. Actually, the diurnal variations are not smooth at each single site.

*Why the linear models with NDVI perform better than the model with Fc? The author should give explanation more clearly. Eq. 2 and Eq. 3 are almost the same, it does not make sense to take them as two different models.*

**Reply:** The weak correlation between G/Rn and NDVI might be the main reason for the poor performance of the methods with Fc. In addition, the NDVI data was used for calculating the Fc in this study. The spatial resolution of the NDVI data was too coarse to represent the scale of flux measurements. This might explain the weak correlation between G/Rn and NDVI.

In Line 498-504, the sentence "The accuracies of the LC_fc_SE and LC_fc_ST methods, which embed fractional vegetation coverage in the G simulation, were significantly lower than those of the previous three methods." has been revised to "The accuracies of the LC_fc_SE and LC_fc_ST methods, which embed fractional vegetation coverage in the G simulation, were satisfied for monthly simulation, but were significantly lower than those of the previous three methods in simulating daily values. The weak correlation between G/Rn and NDVI might be the main reason for the poor performance of these methods. However, the coarse-resolution NDVI data used in this study are not sufficient to represent the scale of flux measurements, especially for sites with heterogeneous land surface. This might be the main reason for this weak correlation. The application of higher resolution and continuous

vegetation index data series is expected to improve the simulation accuracy of these methods."

The five methods (Eq. 2 to 6) are empirical models based on the correlation between the G and Rn. The term containing the NDVI or Fc in Eq. 3 to 6 could be taken as a whole, which is similar to the coefficient in the Eq. 2. However, these methods were commonly used in different remote sensing evapotranspiration models. For example, the Eq. 2 is applied in the TSEB, ALEXI, DisALEXI, GLEAM, and other RS ET models to simulate G, but different models use different linear coefficient values. The Eq. 3 is applied in the SEBAL model. The Eq. 4 is applied in modified SEBAL (Singh et al., 2008) and SEBS (Chen et al., 2019) evapotranspiration models. Eq. 5 and 6 are applied in the S-SEBI, NTSG, BESS, METRIC, MOD16A2, and SEB-4S models. Therefore, Eq 2 to 6 are taken as different methods and evaluated in this study.

*Indeed, the different performance of G estimation in different time (hours of a day) are closely related to the time-lag between G and Rn, which is important but not well explained.*

**Reply:** Yes, the time-lag between G and Rn could be found in Figure 1. The temporal performance of G estimation in each half-hour of a day was shown in Figure 6. Figure 3 shows intra-day distribution of the linear fitted relationship between G and Rn. It is obvious from the Figure 3 and 6 that the performance of G estimation in each half-hour is closely related to the correlation between G and Rn. However, the author has not found the obvious relationship between the performance and the time-lag from the Figure 1 and 6.

*At large scale application using satellite data (NDVI), the author has used the NDVI from AVHRR product with the spatial resolution of 0.05×0.05 °which is too coarse to compare with the footprint of ground measurements. This is particularly important for sites where the land surface is heterogeneous around the 0.05×0.05 °spatial domain.*

**Reply:** Thanks very much for your valuable comments. Yes, the spatial resolution of the NDVI data was too coarse to represent the scale of flux measurements. Several new sentences have been added as follows,

In Line 501-504, "However, the coarse-resolution NDVI data used in this study are not sufficient to represent the scale of flux measurements, especially for sites with

heterogeneous land surface. This might be the main reason for this weak correlation. The application of higher resolution and continuous vegetation index data series is expected to improve the simulation accuracy of these methods."

In Line 518-519, "The poor performance might be mainly caused by the coarse-resolution vegetation index data, which could not represent the scale of flux measurements."

*In the Discussion section the author state that "it requires intra-day land surface temperature (LST) data series, which cannot be obtained by RS. Because RS can only monitor instantaneous LST when a satellite overpasses, it cannot obtain intra-day LST data series", this is not true! Geostationary satellites can provide LST observations at 15min – 30min intervals.*

**Reply:** This sentence "However, it requires intra-day land surface temperature (LST) data series, which cannot be obtained by RS. Because RS can only monitor instantaneous LST when a satellite overpasses, it cannot obtain intra-day LST data series." has been deleted.

*Minor issues:*

*In Figure 1, explanations for sub-figures are missing in the caption.*

**Reply:** Explanations for sub-figures had been added in the caption of Figure 1, as follows, "a is raw Rn, H, LE, G, and G'; b-f are normalized Rn, H, LE, G, and G', respectively"

*Figure 8: I do not understand how to get the median NSE value of each site. Shouldn't it be a single NSE value per site?*

**Reply:** It is the median NSE of 48 half-hour values in diurnal period. The caption of Figure 8 has been revised to "…… a is the median NSE of 48 half-hour values; b-e represent the NSE values at 6:30, 10:30, 13:30 and 18:00, respectively……"

*It's inappropriate to define 6:00-7:00 as sunrise periods and 17:00-18:00 as sunset periods globally, since the sunrise and sunset time vary with locations and seasons.*

**Reply:** According to your valuable comments, the definitions of sunrise and sunset periods have been deleted in the revised manuscript.

*Line 190: "During data processing, data points with absolute values greater than 10 in the G/Rn or G/H daily series of each period were deleted." Threshold 10 seems too large, can G be larger than Rn ?*

**Reply:** There were several data points with absolute values greater than 10 in the observed G/Rn or G/H daily series. For example, in the daily series of observed soil heat flux (G') and net radiation (Rn) at the AR-SLu site, the value of G'/Rn is greater than 10 in 2010-6-18 and 2010-9-3, and is lower than -10 in 2010-5-15, 2010-6-26, 2010-8-3 and 2010-9-27. The value of G'/Rn is larger than 10 in 11 days, and is lower than -10 in 17 days at the AT-Neu site. Similar values could also found in other flux sites. The author could provide the raw data if it is possible.

*Similar problem is in the range of coefficient α in Eq.1, the maximum value of 1.5 will lead to G is 1.5 times larger than Rn, does it have physical meaning?*

**Reply:** The ratio of G and Rn is generally lower than 1.0 for the mean value at observed sites. However, as described above, the daily values of G/Rn might be larger than 1.0. This can happen. For example, the observed Rn and G' is 0.15 $W/m^2$ and 5.0 $W/m^2$ in 2002-10-14. The simulated G on the same day is -15.7 $W/m^2$. The absolute values of G/Rn and G'/Rn are larger than 1.5.

*In Figure 4, analysis is also done for monthly temporal scale without explaining why it is needed.*

**Reply: Figure 4-d** and **4-e** show the linear and exponential fitted $R^2$ values between the monthly series of G/Rn and NDVI at each observed site, respectively. The empirical correlation between G/Rn and NDVI was also analyzed in Figure 4. The results showed that the correlation between G/Rn and NDVI was weak in the daily series but strong in the monthly series. The last paragraph of the section 3.2 (Line 283-301) is the description of these sub-figures. In addition, the revision was also made in Line 498-500.

---

## Author Response (AR3)

**Responses to the Editor's Comments**

**Ms. Ref. No.**: hess-2022-125

**Title**: Accuracy of five ground heat flux empirical simulation methods in the surface energy balance-based remote sensing evapotranspiration models

**Author(s):** *Zhaofei Liu*

**\*\*\*\*\*\*\*\*\*\*\*\*\*\*\*\*\*\*\*\*\*\*\*\*\*\*\*\*\*\*\*\*\*\*\*\*\*\*\*\*\*\*\*\*\*\*\*\*\*\*\*\*\*\*\*\*\*\*\*\*\***

*Comments to the author:*

*Dear author,*

*Thank you very much for submitting your revision and responses to reviewer's comments.*

*I have consulted the reviewers again and would suggest that you consider the following issues:*

*1) The comments raised by Reviewer #4 are important issues. Because you can not completely solve these issues, a practical solution is that you should fully discuss them in the discussion part to inform the reader about the shortcoming of your study.*

*The reviewer suggested that the author does not seem to have assessed and corrected the energy imbalance at the flux tower sites before using the data, which could be the reason that the author has obtained very large G based on the SEBR method at site level and thus would derive unreliable relationships between G and Rn. As such the uncertainty of G estimated by SEBR method would be very large, even larger than the magnitude of G itself.*

*2) With reference to Purdy et al. (2016) who evaluated six empirical methods of G simulation against G' observations at 88 flux sites.*

*The reviewer suggested that the author should contact Purdy to make sure whether they have used G' or G observations at the 88 flux sites.*

*I hope you can take these comments in consideration in your revision.*

*Best wishes*

*Bob Su*

*Editor HESS*

\*\*\*\*\*\*\*\*\*\*\*\*\*\*\*\*\*\*\*\*\*\*\*\*\*\*\*\*\*\*\*\*\*\*\*\*\*\*\*\*\*\*\*\*\*\*\*\*\*\*\*\*\*\*\*\*\*\*\*\*\*\*\*\*\*\*\*\*\*\*\*\*\*\*\*\*\*\*

Dear Prof. Su,

Thanks very much for your valuable comments and suggestion. For your convenience to re-review the paper, the response corresponding to your comments are described in detail as follows:

**Reply 1):** Thanks very much for your valuable comments. A new section "4.1 Limitations and uncertainties" has been added to fully discuss the shortcoming of this study. It includes three points: (1), the uncertainties of the SEBR method used in this study were discussed. (2), the coarse-resolution NDVI data used in this study were discussed. (3), the limited research implications were also identified. The new section is as follows,

**"4.1 Limitations and uncertainties**

Theoretically, surface energy is balanced. The energy unclosure might be mainly caused by the error of the observed data. The observed G' instead of G was generally used to investigate the energy balance ratio (Wilson et al., 2002). The energy balance closure problem might be largely caused by the soil heat storage (Foken, 2008). Compared with G, other energy terms can be observed more accurately. The eddy covariance measurements of H and LE are generally considered to be the most accurate observations available, and have been widely used as a reference to evaluate other simulation methods. The **Eq. (1)** makes full use of the surface energy term that can be accurately measured at present. In other words, it assumes that the measurements of Rn, H and LE are accurate in this study. The uncertainties of measurements are not considered in this study. However, the uncertainty of G

estimated by the SEBR method could be very large at some sites, which have low magnitude of G. Although the majority of sites have G values greater than 10 W/m$^2$, and take up more than 15% of Rn, there are some sites (20/230) with G values lower than 5 W/m$^2$. There would be great uncertainties of the SEBR method in simulating G values at these sites.

The accuracies of the LC_fc_SE and LC_fc_ST methods, which embed fractional vegetation coverage in the G simulation, were satisfied for monthly simulation, but were significantly lower than those of the other three methods in simulating daily values. The weak correlation between G/Rn and NDVI might be the main reason for the poor performance of these methods. However, the coarse-resolution NDVI data used in this study are not sufficient to represent the scale of flux measurements, especially for sites with heterogeneous land surface. This might be the main reason for this weak correlation. The application of higher resolution and continuous vegetation index data series is expected to improve the simulation accuracy of these methods. A large error in the G simulation might be induced in the ET modelling process, thereby reducing the accuracy of the ET estimates. In RS ET models, Rn is generally calculated using radiation balance with RS images and meteorological inputs. However, observed Rn was used for simulating G in this study. In other words, it was assumed that Rn is accurately simulated by the RS ET models. Therefore, it should be noted that the uncertainty in Rn calculation was also a source of error in G simulations in ET models.

The evaluation results of this study are expected to provide reference for RS ET model application and developers. For example, the performance of these methods was good and poor at some sites and time periods and some land-cover types. RS ET modelers could check the advantage of the models at good performance regions, and find why the models are poor at some other areas, then revise the models to improve the accuracy at poor performance regions. However, how to improve the model to improve the accuracy needs further research. "

**Reply 2):** The reference is "Purdy, A. J., Fisher, J. B., Goulden, M. L., and Famiglietti, J. S.: Ground heat flux: an analytical review of 6 models evaluated at 88 sites and globally, J. Geophys. Res. Biogeosci., 121, 3045–3059, doi:10.1002/2016JG003591, 2016." Purdy is the corresponding author of the paper. The correspondence email is ajpurdy@uci.edu.

There are 15 papers when search "Purdy AJ" for author names in the Web of Science. From these papers, the author found that Purdy transferred from University of California to the University of San Diego in 2019. However, the corresponding author for these papers is not Purdy. The author tried to contact the co-author of Purdy's latest paper (Wu, et al, 2020, Evaluating three evapotranspiration estimates from model of different complexity over China using the ILAMB benchmarking system, Journal of Hydrology.), and got another email from him (adamjpurdy@gmail.com).

In addition, the author also tried to search Purdy AJ's contact information from the Internet, but did not obtain any valid information.

The author has been contacting Purdy by emails (ajpurdy@uci.edu and adamjpurdy@gmail.com ) every day since October 30. But there was no reply.

As can be seen from the reference (Purdy, et al., 2016), Purdy used the observed G' data. For example,

In the Abstract, it shows that "We provide the largest review of these methods to date (N = 6), evaluating modeled G against measured G from 88 FLUXNET sites."

In the section of Data Sets, it points that "Overall, we used measurements from 88 towers across 11 climates and 10 biomes to evaluate modeled G", and "Data are available from the FLUXNET database (http:///www.fluxdata.org). Despite being the best available collection of globally distributed observations, many locations lack a full year of observations, experience instrument quality degradation, and locate ground heat flux plates and soil thermocouples to calculate storage at different depths (2–15 cm) to measure G."

In addition, a new reference cited in the section 4.1 has been added as follows,

Foken, T.: The energy balance closure problem: An overview, Ecol. Appl., 18, 1351–1367, https://doi.org/10.1890/06-0922.1, 2008.

In the Acknowledgments, the order of grant numbers "XDA23090302; XDA2006020202" has been revised into "XDA2006020202; XDA23090302".

Best Regards

Zhaofei Liu